# *Cis* and *trans* RET signaling control the survival and central projection growth of rapidly adapting mechanoreceptors

**Michael S Fleming[1†], Anna Vysochan[1†], Sónia Paixão[2], Jingwen Niu[1], Rüdiger Klein[2], Joseph M Savitt[3], Wenqin Luo[1]\***

[1]Department of Neuroscience, Perelman School of Medicine, University of Pennsylvania, Philadelphia, United States; [2]Molecules - Signals - Development, Max Planck Institute of Neurobiology, Martinsried, Germany; [3]Parkinson's Disease and Movement Disorder Center of Maryland, Elkridge, United States

**Abstract** RET can be activated in *cis* or *trans* by its co-receptors and ligands in vitro, but the physiological roles of *trans* signaling are unclear. Rapidly adapting (RA) mechanoreceptors in dorsal root ganglia (DRGs) express *Ret* and the co-receptor *Gfrα2* and depend on *Ret* for survival and central projection growth. Here, we show that *Ret* and *Gfrα2* null mice display comparable early central projection deficits, but *Gfrα2* null RA mechanoreceptors recover later. Loss of *Gfrα1*, the co-receptor implicated in activating RET *in trans*, causes no significant central projection or cell survival deficit, but *Gfrα1;Gfrα2* double nulls phenocopy *Ret* nulls. Finally, we demonstrate that GFRα1 produced by neighboring DRG neurons activates RET in RA mechanoreceptors. Taken together, our results suggest that *trans* and *cis* RET signaling could function in the same developmental process and that the availability of both forms of activation likely enhances but not diversifies outcomes of RET signaling.

**\*For correspondence:** luow@mail.med.upenn.edu

[†]These authors contributed equally to this work

## Introduction

The neurotrophic receptor tyrosine kinase RET plays critical roles in many biological processes, including kidney genesis, spermatogenesis, and development of enteric, sensory, autonomic, and motor neurons (*Runeberg-Roos and Saarma, 2007*; *Ibáñez, 2013*). Loss of RET signaling leads to Hirschprung's disease, while RET gain of function has been implicated in various human carcinomas (*Runeberg-Roos and Saarma, 2007*; *Santoro and Carlomagno, 2013*). In addition, activation of the RET signaling pathway has potential applications in the treatment of Parkinson's disease and promotion of spinal cord (SC) regeneration following injury (*Bespalov and Saarma, 2007*; *Deng et al., 2013*). Therefore, it is critical to thoroughly understand RET signaling mechanisms.

RET is the common signaling receptor for the glial cell line-derived neurotrophic factor (GDNF) family of ligands (GFLs), which includes GDNF, neurturin (NRTN), artemin, and persephin. For RET activation and signaling, GFLs first bind to a GPI-linked GDNF family receptor alpha (GFRa), which then associates with RET to form an active signaling complex (*Airaksinen and Saarma, 2002*). In vertebrates, the GFRas and their high-affinity ligand pairs are GFRa1 and GDNF (*Jing et al., 1996*; *Treanor et al., 1996*), GFRa2 and NRTN (*Baloh et al., 1997*; *Buj-Bello et al., 1997*; *Klein et al., 1997*), GFRa3 and artemin (*Baloh et al., 1998*), and GFRa4 and persephin (*Yang et al., 2007*).

RET can be activated by GFRas expressed in the same cell (*cis* signaling) or by GFRas (mainly GFRa1) produced from other sources (*trans* signaling) in vitro (*Paratcha et al., 2001*; *Ledda et al., 2002*). The existence of both *cis* and *trans* activation has been proposed to diversify RET signaling by either recruiting different downstream effectors or changing the kinetics or efficacy of kinase activation

**eLife digest** During development, cells send and receive numerous signaling molecules. In order to trigger a biological response, such signaling molecules must first bind to a specific receptor protein, often located on the cell surface. These receptor proteins can either work alone or with partner proteins called co-receptors. When the co-receptor is produced by the same cell as the receptor, it is called *cis* signaling. When the co-receptor is produced by other cells, it is called *trans* signaling.

RET is one such receptor that is important for the development of the nervous system and many other biological processes. It interacts with a particular family of signaling molecules, the glial cell line-derived neurotrophic factor (GDNF) family ligands, which first bind to a co-receptor, GFRα, before binding to RET. These co-receptors can come from the same cell as RET, or from a different cell.

Previous studies have indicated that RET can receive both *cis* and *trans* signals using cultured cells, but it was not clear whether both types of signal occur during normal development and contribute to the same biological processes. Fleming, Vysochan et al. investigated this question by analyzing the roles of RET signaling in a type of mouse neuron that is involved in sensing touch. RET is important for the survival and development of these neurons, which express both RET and its co-receptor GFRa2. Another RET co-receptor, GFRa1, is produced by other cells that are next to the cell bodies and projections of these touch-sensing neurons.

To investigate the roles of different GFRa co-receptors further, Fleming, Vysochan et al. generated a variety of mouse mutants, including mice with mutations in one or both types of co-receptor. The neurons in mice lacking both co-receptors shared the same defects as the neurons in the mice lacking RET. Loss of either co-receptor alone did not produce these abnormalities. This indicates that both co-receptors can mediate the normal development of these neurons, with GFRa2 signaling in *cis* and GFRa1 signaling in *trans*.

Fleming, Vysochan et al. propose that *cis* and *trans* RET signaling can lead to the same biological outcomes in these neurons. Future experiments should reveal if *cis* and *trans* RET signaling contribute towards common biological processes in other cell types inside the body as well. Such findings might also be important for understanding the role of RET signaling in cancer and other human diseases.

---

(*Tansey et al., 2000*; *Paratcha et al., 2001*). Consistent with the *trans* signaling model, *Gfra1* is expressed in the target fields of many RET⁺ neurons during development and can promote axon growth upon GDNF treatment in culture (*Trupp et al., 1997*; *Yu et al., 1998*; *Paratcha et al., 2001*). However, the 'cis-only' mouse model, in which *Gfra1* is expressed under the control of the *Ret* promoter in a *Gfra1* null background, produced no overt phenotypes in many *Ret*-dependent developmental processes, suggesting that *trans* signaling may not play a major physiological role (*Enomoto et al., 2004*). Recently, *trans* RET signaling has been implicated in the development of inhibitory cortical interneurons, nigral dopaminergic neurons, and enteric lymphoids, and in perineural invasion by cancer cells (*Canty et al., 2009*; *Kholodilov et al., 2011*; *Patel et al., 2012*; *He et al., 2014*). Nevertheless, the physiological functions of *trans* RET signaling and whether *cis* and *trans* signaling lead to the same or different biological outcomes in vivo remain largely unresolved.

Aβ mechanoreceptors are large diameter somatosensory neurons mediating discriminative touch, which innervate layers III–V of the SC. They can be broadly divided into rapidly adapting (RA) and slowly adapting (SA) mechanoreceptors based on their adaptation properties to sustained mechanical stimuli (*Fleming and Luo, 2013*). Previously, we and other labs identified that a small population of mouse DRG (dorsal root ganglion) neurons, the early RET⁺ DRG neurons, develop into RA mechanoreceptors, and that *Ret* is required cell autonomously for the growth of their third order central projections innervating the dorsal SC (dSC) (*Bourane et al., 2009*; *Luo et al., 2009*; *Honma et al., 2010*).

RET in RA mechanoreceptors encounters environments in which both *cis* and *trans* activation are possible, providing a good model system to study the physiological functions of *trans* RET signaling. RA mechanoreceptors express *Ret* and *Gfra2* (*Bourane et al., 2009*; *Luo et al., 2009*; *Honma et al., 2010*), whereas *Gfra1* is highly expressed in their target field (*Treanor et al., 1996*; *Yu et al., 1998*) and by

neighboring DRG neurons during development (*Luo et al., 2009*; *Honma et al., 2010*). Here, we found that the central projection deficit of RA mechanoreceptors is negligible in postnatal *Gfra2* and *Nrtn* mutant mice, which is in great contrast to the severely affected *Ret* mutant mice. By genetically tracing RA mechanoreceptors in different mutant mouse lines during development, we showed that the initial growth of the third order central projections of RA mechanoreceptors depends on the *cis* activation of RET via GFRa2 and NRTN. However, central projections of *Gfra2* null RA mechanoreceptors gradually recover during development. *Gfra1* null mice show no obvious central projection deficit by itself, but *Gfra1;Gfra2* double null mice have similar cell death and central projection deficits to those of *Ret* null mice. Moreover, we showed that *Gfra1* is non-detectable in most RA mechanoreceptors, thus RET in RA mechanoreceptors is most likely activated by GFRa1 in *trans*. Finally, we determined that RET in *Gfra2* null RA mechanoreceptors responds to GDNF in DRG explant culture, and this responsiveness is mediated by GFRa1 from neighboring DRG neurons (*trans* activation). Taken together, our results indicate that combinatorial *cis* and *trans* RET signaling promote survival and central projection growth of RA mechanoreceptors in vivo.

## Results

### Expression of *Ret*, *Gfras*, and *GFLs* in the developing mouse SC and DRGs

Since RET can be activated by GFLs/GFRas either in *cis* or in *trans* (mainly by GDNF/GFRa1) in vitro, we asked if the expression patterns of *Gfra1*, *Gfra2*, *Gdnf*, and *Nrtn* in the developing SC and DRGs would provide insight into RET signaling in RA mechanoreceptors in vivo. We performed in situ hybridization for *Ret*, *Gfra1*, *Gfra2*, *Gdnf*, and *Nrtn* on embryonic day 13.5 (E13.5) and E15.5 wild-type DRG and SC sections. Double in situ hybridizations that characterize the expression of *Gfra1* and *Gfra2* in different populations of DRG neurons have been previously conducted (summarized in *Figure 1—figure supplement 1K* [*Luo et al., 2009*]).

Similar to previous characterization (*Molliver et al., 1997*; *Luo et al., 2007*, *2009*), *Ret* is expressed in motor neurons and a mix of small and large diameter DRG neurons at E13.5 and E15.5 (*Figure 1—figure supplement 1A–B*). Most large diameter RET[+] DRG neurons at these stages are the early RET[+] DRG neurons, which develop into RA mechanoreceptors (*Bourane et al., 2009*; *Luo et al., 2009*). *Gfra1* is highly expressed in some DRG neurons and motor neurons as well, but these GFRa1[+] DRG neurons come from NTRK1[+] precursors and are not early RET[+] RA mechanoreceptors (*Yu et al., 1998*; *Luo et al., 2009*; *Honma et al., 2010*). In addition, *Gfra1* is highly expressed in the dorsal root entry zone and the dSC, which are the target fields of the central projections of RA mechanoreceptors (*Figure 1—figure supplement 1C–D*). *Gfra2* is expressed in a small number of large diameter DRG neurons, which were previously shown to be RA mechanoreceptors (*Bourane et al., 2009*; *Luo et al., 2009*), and some SC cells and motor neurons at these stages (*Figure 1—figure supplement 1E–F*; *Oppenheim et al., 2000*).

*Nrtn* is diffusely expressed at a low level in the SC and DRGs at both E13.5 and E15.5; *Gdnf* transcript is barely detected at E13.5 but is clearly expressed in DRG and motor neurons at E15.5 (*Figure 1—figure supplement 1G–J*). Thus, based on the expression patterns of RET signaling components in the developing SC and DRGs, RET in the central projections and cell bodies of developing RA mechanoreceptors could potentially be activated in *cis* by NRTN/GFRa2 or in *trans* by GDNF/GFRa1, which may come from neighboring DRG neurons, dorsal root entry zone cells, or dSC cells.

### Central projection deficit of RA mechanoreceptors is negligible in postnatal *Gfra2* and *Nrtn* null mice

RA mechanoreceptors depend on RET for the growth of their third order central projections innervating layers III–V of SC. In postnatal *Ret* mutant mice, VGLUT1[+] puncta, which label pre-synaptic terminals of mechanoreceptors and proprioceptors (*Hughes et al., 2004*; *Paixão et al., 2013*), are greatly reduced in layers III–V, indicating deficits in the third order central projections of RA mechanoreceptors (*Luo et al., 2009*). Since RA mechanoreceptors express a high level of *Gfra2* but not any other *Gfras* (*Luo et al., 2009*), it is likely that RET in RA mechanoreceptors is activated by NRTN/GFRa2 in *cis*. Indeed, we previously found that Pacinian corpuscles, a subtype of RA mechanosensory end organs in the periphery, are not formed in *Ret*, *Gfra2*, or *Nrtn* mutant mice, supporting that NRTN/GFRa2-RET *cis* signaling occurs in RA mechanoreceptors (*Luo et al., 2009*). Here, we asked if NRTN-GFRa2/RET *cis*

signaling is required for the growth of RA mechanosensory central projections as well. We performed immunostaining of VGLUT1 with postnatal day 7 (P7) $Gfra2^{GFP/GFP}$ null and $Nrtn^{-/-}$ null SC sections. No significant decrease of VGLUT1$^+$ puncta in layers III–V of SC is observed in $Gfra2$ and $Nrtn$ null mice (*Figure 1A–C*, *Figure 1—source data 1* [p = 0.96], and data not shown). This result suggests that unlike RET signaling in the peripheral branches of RA mechanoreceptors, *cis* activation of RET by GFRa2 and NRTN may be dispensable for the normal development of central projections of RA mechanoreceptors.

To determine whether RA mechanoreceptors survive without $Gfra2$, we quantified the number of GFP$^+$;NF200$^+$ neurons per DRG section in P7 $Gfra2^{GFP/+}$ controls and $Gfra2^{GFP/GFP}$ nulls. Green fluorescent protein (GFP) is expressed from the $Gfra2$ locus and most of GFP$^+$;NF200$^+$ neurons indicate RA mechanoreceptors in $Gfra2^{GFP}$ mice. In agreement with our previous findings at P0 (*Luo et al., 2009*), we found a slight but non-significant decrease in RA mechanoreceptor number between controls and mutants (*Figure 1D*, *Figure 1—source data 1* [p = 0.34]). Therefore, *cis* RET signaling via GFRa2 does not seem to be critical for the early postnatal survival of RA mechanoreceptors.

## Central projection deficit of *Ret* null mice at E13.5

To understand the mechanism of RET signaling that controls growth of RA mechanosensory central projections, we genetically traced RA mechanoreceptors in *Ret*, *Gfra1*, *Gfra2*, and *Nrtn* mutant mice at different developmental stages. We first used *Ret* mutant mice, which serve as a positive control for the central projection deficit, to determine a robust method for visualizing RA mechanosensory interstitial branches at E13.5. We compared two methods that have been previously used. One is immunostaining of neurofilament-200 (NF200), which is expressed by large diameter DRG neurons, including RA mechanoreceptors, SA mechanoreceptors, and proprioceptors (*Bourane et al., 2009*). The other is to use a knockin/null allele of *Ret* (*Honma et al., 2010*), $Ret^{CFP}$ (*Uesaka et al., 2008*), in which cyan fluorescent protein (CFP, a variant of GFP) is expressed from the *Ret* locus. Although *Ret* is expressed in both RA mechanoreceptors and some other DRG neurons at E13.5 (*Luo et al., 2009*), central projections of non-RA mechanoreceptor RET$^+$ neurons, most of which develop into nociceptors, do not innervate the dSC until E15.5 or later (*Ozaki and Snider, 1997*). In addition, the expression of *Ret* in dSC neurons is not obvious until E15.5 (*Figure 1—figure supplement 1B*). Thus, the $Ret^{CFP}$ allele may allow us to specifically visualize central projections of RA mechanoreceptors at E13.5.

To compare these two methods, we performed anti-NF200 and anti-GFP immunostaining on SC sections of E13.5 $Ret^{CFP/+}$ control and $Ret^{CFP/CFP}$ null embryos. We observed a decrease in the density of NF200$^+$ fibers in the dorsal horn (*Figure 1—figure supplement 2B,E*). This decrease of NF200$^+$ central projections, however, is not dramatic. This is because the NF200 antibody also recognizes central projections of SA mechanoreceptors and proprioceptors, which develop in a manner temporally comparable to RA mechanoreceptors. In contrast, CFP$^+$ fibers innervating the dSC display a dramatic reduction in *Ret* null mice (*Figure 1—figure supplement 2C,F*). *Ret* null CFP$^+$ fibers reach the dorsal surface of the SC but rarely grow interstitial branches innervating layers III–V. We quantified the number of CFP$^+$ pixels in the dorsal horn (displayed as percentage of CFP$^+$ pixels normalized to the control) as a proxy for the extent of axon growth and found a significant decrease in CFP$^+$ fibers in *Ret* mutant dorsal horn (*Figure 1—figure supplement 2H* and *Figure 1—source data 2*, [p < 0.001]). This result suggests that the $Ret^{CFP}$ allele is a valid tool for visualizing central projection deficits of RA mechanoreceptors at E13.5.

Since *Ret* signaling can positively regulate the expression of its own signaling components or control neuronal survival (*Luo et al., 2007*; *Baudet et al., 2008*; *Golden et al., 2010*), it is conceivable that the lack of dSC CFP$^+$ fibers could be due to a downregulation of *CFP* expressed in *Ret* null RA mechanoreceptors or death of RA mechanoreceptors. To exclude these possibilities, we quantified the number of CFP$^+$ neurons in DRGs. We found that the number of CFP$^+$ neurons per DRG section was not statistically different between *Ret* heterozygotes and null mice (*Figure 1—figure supplement 2I* and *Figure 1—source data 2*). In addition, the intensity of GFP$^+$ fibers at the dorsal surface of the SC is comparable between *Ret* mutant and control mice. Therefore, the loss of CFP$^+$ fibers in the dorsal horn of E13.5 *Ret* mutants must mainly be due to a deficit in growth of interstitial central axons, but not due to the down-regulation of *CFP* expression or the death of RA mechanoreceptors.

## Central projections of RA mechanoreceptors are normal in E13.5 *Gfra1* null mice

The finding that dSC VGLUT1 staining is largely normal in postnatal *Gfra2* and *Nrtn* null mice suggests that *cis* RET signaling may be dispensable for RA mechanosensory central projections.

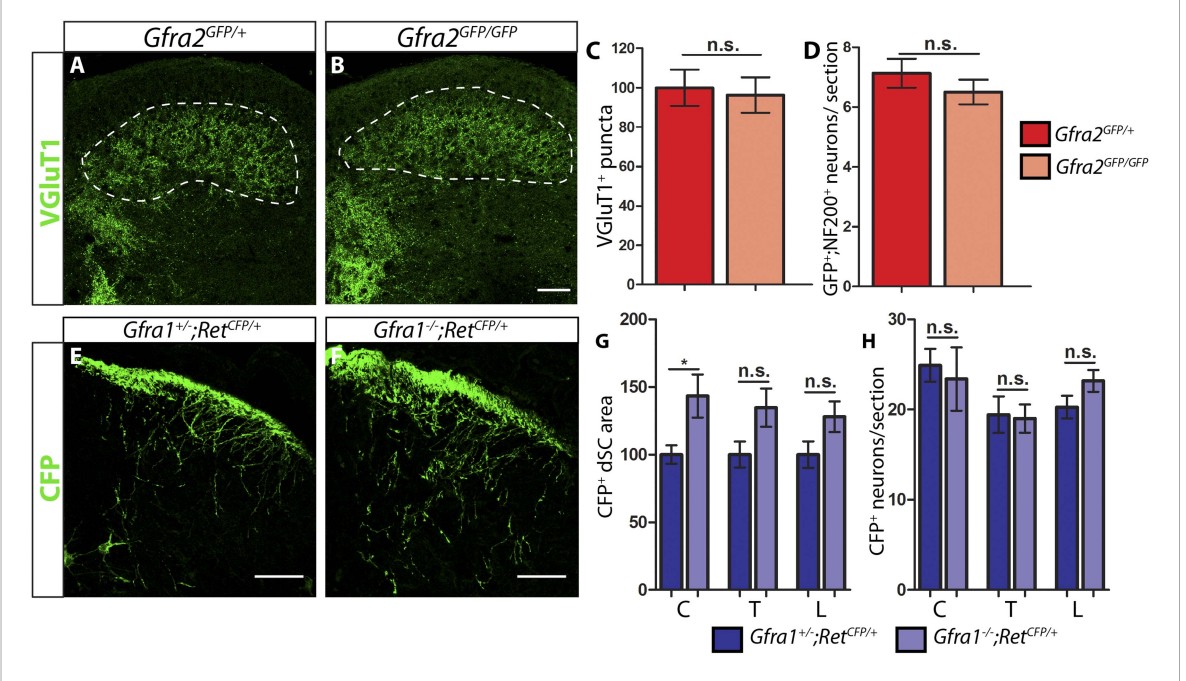

**Figure 1**. P7 *Gfra2* mutant mice show normal dorsal spinal cord (dSC) VGLUT1 staining and *Gfra1* null mice display normal rapidly adapting (RA) mechanoreceptor central projections at E13.5. (**A–B**) Anti-VGLUT1 immunostaining of P7 SC sections from *Gfra2GFP/+* control (**A**) and *Gfra2GFP/GFP* null (**B**) mice. VGLUT1 staining labels presynaptic terminals of mechanosensory neurons, which are found in layers III–V of the dSC (outlined in white). Note that green fluorescent protein (GFP) driven from the *Gfra2* locus cannot be visualized directly. Therefore, positive signal indicates presynaptic VGLUT1+ puncta and not GFRa2+ primary afferent axons. (**C**) Quantification of VGLUT1+ puncta in dSC, which is displayed as a percentage of VGLUT1+ pixels compared to the control pixel count. The similar density of VGLUT1+ puncta between mutant and control tissue suggests that *cis* RET signaling via GFRa2 is dispensable for the growth of RA mechanosensory central projections at P7. (**D**) Quantification of GFP+;NF200+ neurons, which indicate RA mechanoreceptors, per DRG section. The non-significant decrease in RA mechanoreceptor number per section in *Gfra2* nulls suggests that most RA mechanoreceptors are not dependent on *cis* RET signaling for survival. (**E–F**) Anti-GFP immunostaining of RA mechanoreceptor central projections in E13.5 *Gfra1+/−;RetCFP/+* control (**E**) and *Gfra1−/−;RetCFP/+* mutant (**F**) SC sections. The increased CFP signal in *Gfra1* null dSC is likely due to the precocious expression of *Ret* in some dSC neurons of *Gfra1* mutants. (**G**) Quantification of CFP+ pixel number in dSC. The lack of a reduction in CFP+ axons in *Gfra1* mutant dSC indicates that *trans* signaling via GFRa1 is not required for the initial growth of RA mechanosensory third order central projections. (**H**) Quantification of number of CFP+ neurons per DRG section indicates no loss of RA mechanoreceptors in *Gfra1* mutants at E13.5. C: cervical level, T: thoracic level, L: lumbar level. Scale bars = 50 μm. Error bars represent SEM. n.s. = p > 0.05, * = p < 0.05. Source data are provided in *Figure 1—source data 1, 2*.

The following source data and figure supplements are available for figure 1:

**Source data 1**. VGLUT1 dSC staining and RA mechanoreceptor number in P7 *Gfra2* mutants.

**Source data 2**. RA mechanoreceptor central projections and cell number in E13.5 *Ret, Gfra1, Gfra2,* and *Nrtn* mutants.

**Figure supplement 1**. Expression of *Ret, Gfras,* and *Gfls* in developing spinal cord (SC) and DRG.

**Figure supplement 2**. *Ret* is required for the growth of RA mechanosensory third order central projections at E13.5.

**Figure supplement 3**. Generation of *Gfra1* conditional and null alleles.

To determine if the development of RA mechanosensory central projections depends on the *trans* activation of RET via GFRa1 and GDNF, we generated *Gfra1* null (*Gfra1−*) mice (*Figure 1—figure supplement 3A–B* and 'Materials and methods'). In situ hybridization of *Gfra1* control and null DRG sections showed that *Gfra1* transcripts are not produced in mice homozygous for this mutant allele (*Figure 1—figure supplement 3C–D*). In addition, kidneys are not formed in these *Gfra1* null mice (data not shown), a phenotype consistent with previously reported *Gfra1* null mice (*Cacalano et al., 1998*; *Enomoto et al., 1998*). Thus, the *Gfra1−* allele we generated is a null allele.

If *trans* activation of RET via GFRa1 is required for the growth of interstitial central projections of RA mechanoreceptors, we expect to see a decrease of central projections of RA mechanoreceptors in the dSC of *Gfra1* null mice. To test this idea, we generated E13.5 *Gfra1*$^{+/-}$;*Ret*$^{CFP/+}$ control and *Gfra1*$^{-/-}$; *Ret*$^{CFP/+}$ mutant embryos to examine RA mechanosensory central projections at this stage (*Figure 1E–F*). We found that innervation of dSC by CFP$^+$ fibers was not reduced upon *Gfra1* ablation (*Figure 1G*, *Figure 1—source data 2*). Additionally, the lack of *Gfra1* function did not lead to a decrease of CFP$^+$ DRG neurons (*Figure 1H*, *Figure 1—source data 2*). Together, our results suggest that *trans* activation of RET via GFRa1 is not required for the survival or central projection growth of RA mechanosensory neurons at E13.5.

## *Gfra2* and *Nrtn* mutant mice phenocopy central projection deficits of *Ret* mutant mice at E13.5

Since no deficit was observed in the central projections of RA mechanoreceptors in E13.5 *Gfra1* mutants, we next asked whether *cis* RET signaling is required for the initial growth of RA mechanosensory third order central projections. We crossed the *Ret*$^{CFP}$ allele into *Gfra2* and *Nrtn* null mice and examined central projections of RA mechanoreceptors at E13.5 (*Figure 2A–D*). In contrast to what we observed at P7, at this early development stage CFP$^+$ central projections of RA mechanoreceptors are greatly reduced in both *Gfra2* and *Nrtn* null SC sections (*Figure 2E–F*, *Figure 1—source data 2*, *Gfra2* mutant has 9.50 ± 1.44% of control staining at thoracic levels [p < 0.001]). In addition, similar to the E13.5 *Ret* mutant mice, the number of CFP$^+$ DRG neurons in *Gfra2* and *Nrtn* null mice is comparable to that of control mice (*Figure 2G–H*, *Figure 1—source data 2*), suggesting that the loss of CFP$^+$ fibers in the dSC of these mutant mice is due to a deficit in the interstitial central projection growth of RA mechanoreceptors. Thus, at E13.5, *Gfra2* and *Nrtn* null mice phenocopy the central projection deficit of *Ret* mutant mice, which suggests that RET is activated by NRTN/GFRa2 in *cis* for the initial growth of RA mechanosensory central projections.

## Interstitial central projections of *Gfra2* null RA mechanoreceptors begin to recover from E15.5

If *Ret*, *Gfra2*, and *Nrtn* null mice phenocopy each other at E13.5, why do their postnatal VGLUT1 staining patterns look so different (*Figure 1* and [*Luo et al., 2009*])? One possibility is that since *Ret* has a much broader expression pattern than *Gfra2* in the dSC and DRGs, the dramatic loss of VGLUT1 staining in layers III–V of SC may be caused by the loss of RET signaling both in RA mechanoreceptors and other RET$^+$ cells. For *Gfra2* and *Nrtn* mutant mice, though central projection deficits of RA mechanoreceptors may persist postnatally, VGLUT1$^+$ puncta from SA mechanoreceptors could mask the phenotype. Alternatively, RA mechanosensory central projections in *Gfra2* and *Nrtn* mutant mice could recover at later developmental stages due to the function of other RET signaling mechanisms.

To differentiate these possibilities, we examined central projections of RA mechanoreceptors in *Gfra2* null mice through development. We focused on *Gfra2* instead of *Nrtn* mutant mice because: (1) the cell autonomous requirement of a co-receptor is the key to differentiate *cis* vs *trans* RET signaling; and (2) *Gfra2* and *Nrtn* null mice display very similar phenotypes of RA mechanoreceptors. Since *Ret* begins to be expressed in additional populations of DRG neurons (*Molliver et al., 1997*; *Luo et al., 2007*) and dSC cells (*Figure 1—figure supplement 1B*) from E15.5, we can't use the *Ret*$^{CFP}$ allele to visualize the central projections of RA mechanoreceptors at late developmental stages. To overcome this problem, we used a tandem allele (see 'Materials and methods' and *Figure 3—figure supplement 1*) of an inducible Cre allele of *Ret* (*Ret*$^{CreERT}$) and Rosa26 conditional red fluorescent protein (*Rosa*$^{Tdt}$). We combined these alleles with early (E11.5 and E12.5) 4-hydroxy tamoxifen (4-HT) treatment to specifically trace RA mechanoreceptors, as previously established (*Luo et al., 2009*).

We generated *Gfra2*$^{GFP/+}$; *Ret*$^{CreERT}$; *Rosa*$^{Tdt}$ control and *Gfra2*$^{GFP/GFP}$; *Ret*$^{CreERT}$; *Rosa*$^{Tdt}$ mutant mice and examined their SC and DRG sections at E15.5. Tdt$^+$ fibers innervate layers III–V of the SC, which is consistent with specific genetic tracing of RA mechanoreceptors (*Luo et al., 2009*). In addition, the majority of Tdt$^+$ DRG neurons are RET$^+$, GFRa2$^+$ (reported by the expression of GFP), but NTRK1$^-$ at E15.5 (*Figure 3A–J*), further supporting the specific labeling of RA mechanoreceptors. We found that central projections of *Gfra2* null RA mechanoreceptors are also decreased at E15.5 (*Figure 3K–N*, *Figure 3—source data 1*, *Gfra2* mutant has 55.13 ± 2.82% of control staining at the thoracic level [p < 0.001]). Since the number of labeled DRG neurons is not significantly reduced in the mutant mice (*Gfra2* mutants have 79.52 ± 8.39% of control cell number [p = 0.06]), the central projection

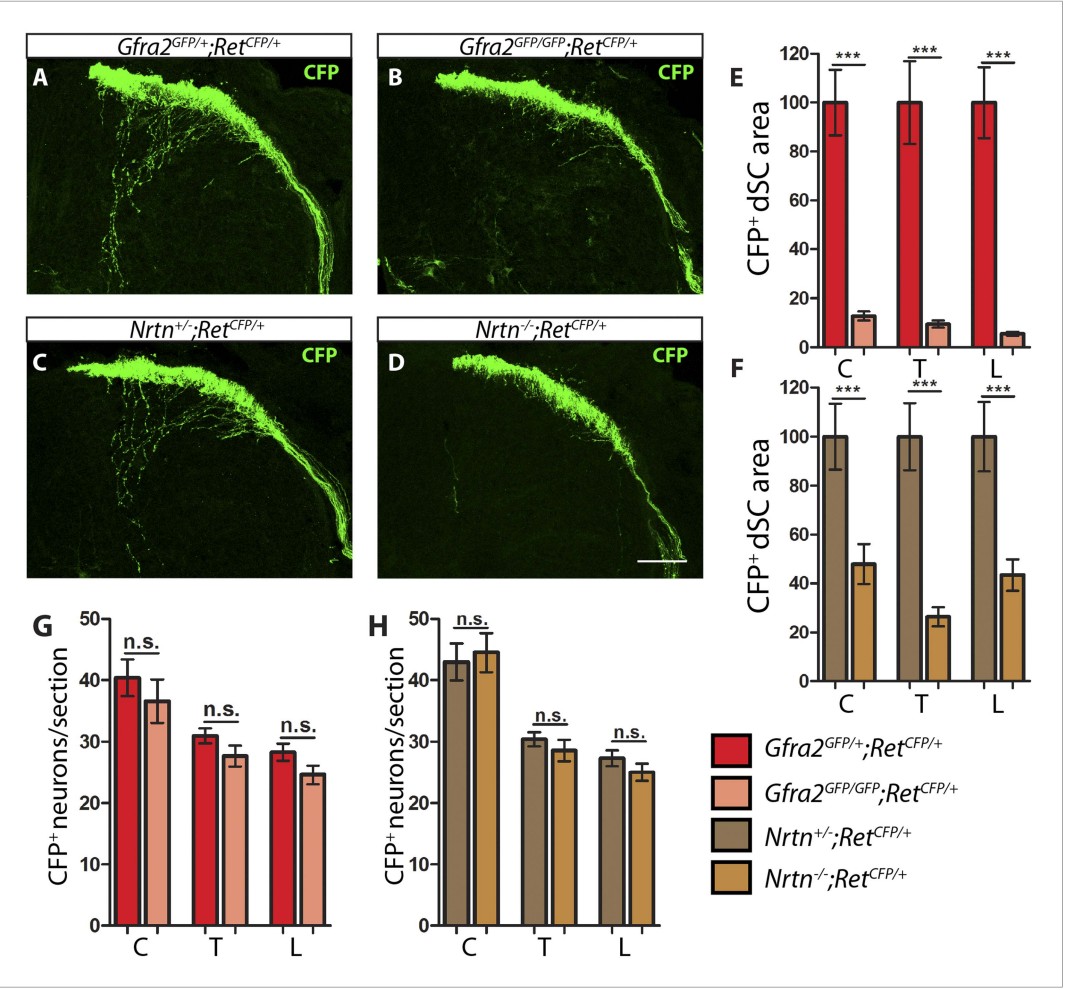

**Figure 2**. *Gfra2* and *Nrtn* null mice show reduced RA mechanoreceptor central projections at E13.5. (**A–D**) Anti-GFP immunostaining to visualize RA mechanosensory central projections in E13.5 dSC sections of *Gfra2*$^{GFP/+}$;*Ret*$^{CFP/+}$ control (**A**), *Gfra2*$^{GFP/GFP}$;*Ret*$^{CFP/+}$ mutant (**B**), *Nrtn*$^{+/−}$;*Ret*$^{CFP/+}$ control (**C**), and *Nrtn*$^{−/−}$;*Ret*$^{CFP/+}$ mutant (**D**) mice. (**E–F**) Quantification of CFP$^+$ pixel number in dSC of *Gfra2* (**E**) and *Nrtn* (**F**) mice. The dramatic reduction in CFP$^+$ axons in *Gfra2* and *Nrtn* nulls at E13.5 suggests that *cis* activation of RET is required for the initial growth of RA mechanosensory third order central projections. (**G–H**) Quantification of number of CFP$^+$ neurons per DRG section in *Gfra2* (**G**) and *Nrtn* (**H**) mice. Similar number of CFP$^+$ DRG neurons between control and mutant mice indicates that cell death of RA mechanoreceptors or downregulation of *Ret*$^{CFP}$ allele do not occur at E13.5 when *cis* RET signaling is ablated. Scale bar = 50 µm. Error bars represent SEM. n.s. = p > 0.05, ** = p < 0.01 *** = p < 0.001. Source data are provided in *Figure 1—source data 2*.

phenotype mostly reflects a growth deficit at this developmental stage. Noticeably, the relative reduction of innervation in *Gfra2* null mice at E15.5 is less severe compared to that of E13.5 mutants (*Figure 2*), suggesting that central projections of *Gfra2* null RA mechanoreceptors may start to recover at this stage.

## *Ret* and *Gfra2* null mice display different central projection and cell survival deficits at E18.5

To determine if RA mechanoreceptors require *Ret* but not *Gfra2* for their central projection growth at later developmental stages, we generated E18.5 *Ret*$^{CreERT/+}$;*Rosa*$^{Tdt}$ control and *Ret*$^{CreERT/CreERT}$;*Rosa*$^{Tdt}$ mutant embryos (*Ret*$^{CreERT}$ is a null allele of *Ret*). Consistent with previous results (*Bourane et al., 2009*; *Luo et al., 2009*; *Honma et al., 2010*), we found that RA mechanosensory central projections are greatly reduced in the *Ret* mutant mice (*Figure 4A,C,I*, *Figure 4—source data 1*, *Ret* mutant has

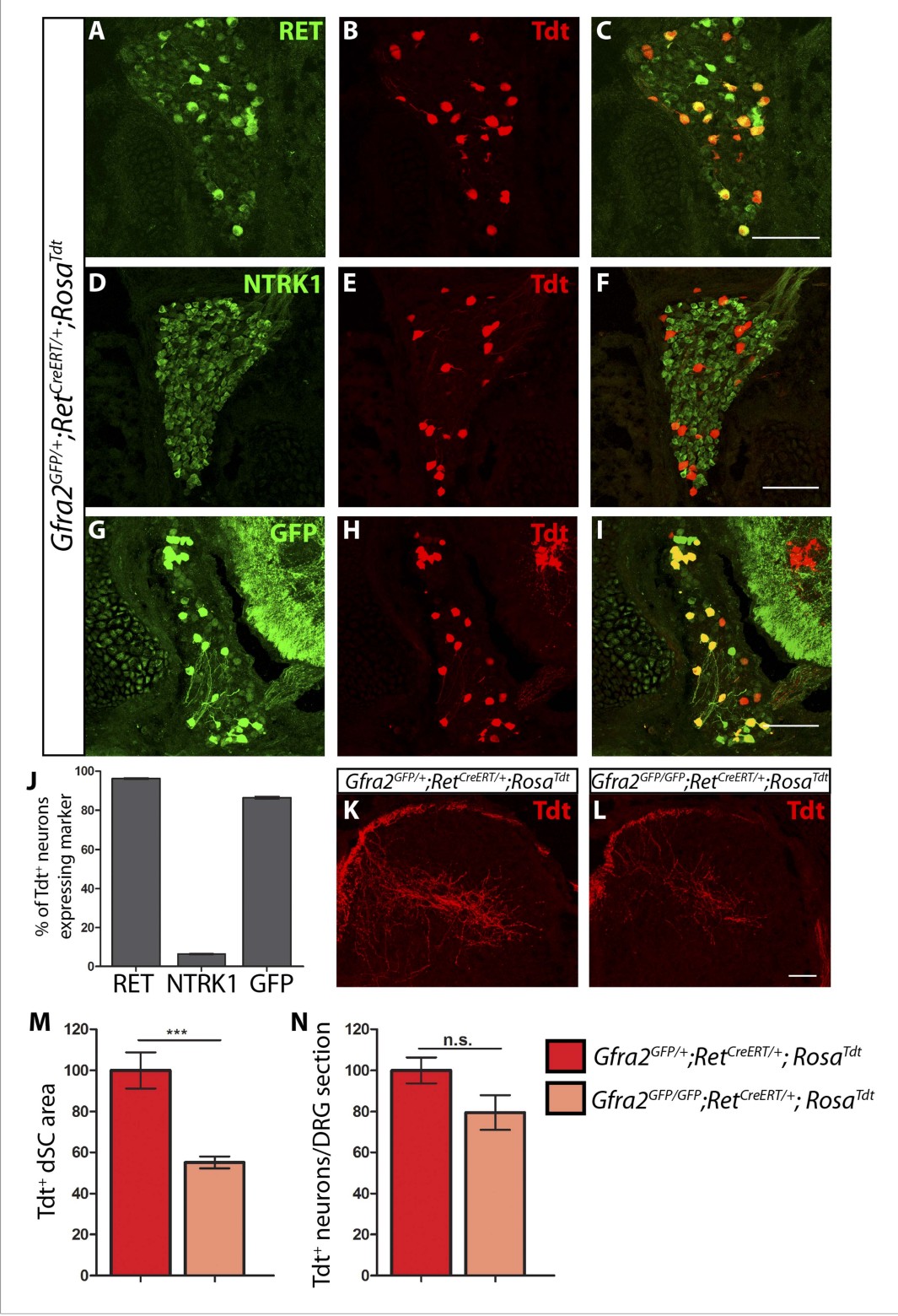

**Figure 3**. Central projection growth deficit of *Gfra2* null RA mechanoreceptors at E15.5. (**A–I**) E15.5 *Gfra2^{GFP/+}*; *Ret^{CreERT/+}*;*Rosa^{Tdt}* DRG sections stained with anti-RET (**A–C**), anti-NTRK1 (**D–F**), and anti-GFP (**G–I**). (**J**) Quantification of percentage of Tdt+ DRG neurons which co-express RET (96.16 ± 0.28%), NTRK1 (6.56 ± 0.18%), and GFP driven from the *Gfra2* locus (86.48 ± 0.55%). The expression profile of Tdt+ neurons confirms that this genetic labeling strategy specifically labels RA mechanoreceptors. (**K–L**) Visualization of Tdt+ RA mechanosensory central projections in dSC of E15.5 *Gfra2^{GFP/+}*; *Ret^{CreERT/+}*; *Rosa^{Tdt}* control (**K**) and *Gfra2^{GFP/GFP}*; *Ret^{CreERT/+}*; *Rosa^{Tdt}* mutant (**L**) SC sections.
*Figure 3. continued on next page*

*Figure 3. Continued*

(**M**) Quantification of Tdt+ pixels in dSC, which is displayed as a percentage normalized to dSC Tdt+ pixels of the within litter controls. *Gfra2* mutant mice have $55.13 \pm 2.82\%$ of control staining (p < 0.001). Note that although *Gfra2* null RA mechanoreceptors still have a central projection deficit at E15.5, the reduction at this stage is less severe than the deficit observed at E13.5. (**N**) Quantification of number of Tdt+ neurons per DRG section, which is displayed as a percentage normalized to Tdt+ neurons of the within litter controls. *Gfra2* mutant mice have $79.52 \pm 8.39\%$ of control cell number (p = 0.06), which suggests that the survival of RA mechanoreceptors is not dependent on *cis* signaling at this stage. Scale bars = 100 μm (**A**–**I**), 50 μm (**K**–**L**). Error bars represent SEM. n.s. = p > 0.05, *** = p < 0.001. Source data are provided in *Figure 3—source data 1*.

The following source data and figure supplement are available for figure 3:

**Source data 1**. RA mechanoreceptor central projections and cell number in E15.5 *Gfra2mutants*.

**Figure supplement 1**. Generation of tandem *Ret^CreERT^;Rosa^Tdt^* allele.

$35.86 \pm 4.97\%$ of control staining at thoracic levels [p < 0.001]). In addition, we counted the number of Tdt+ neurons in L4/L5 DRGs and found that the number of Tdt+ RA mechanoreceptors is dramatically reduced as well (*Figure 4B,D,J*, *Ret* mutant has $52.52 \pm 7.76\%$ of control cell number [p < 0.001]). Taken together, these results suggest that *Ret* is absolutely required for both survival and central projection growth of RA mechanoreceptors at E18.5.

In contrast, central projections of Tdt+ *Gfra2* null RA mechanoreceptors are only slightly reduced at E18.5 (*Figure 4E,G,I*, *Figure 4—source data 1*, *Gfra2* mutant has $86.34 \pm 4.48\%$ of control staining at thoracic levels [p = 0.01]). At P7, almost no difference is observed (data not shown). Similarly, the number of Tdt+ RA mechanoreceptors is only slightly reduced in *Gfra2* null mice (*Figure 4F,H,J Gfra2* mutant has $84.01 \pm 5.16\%$ of control cell number [p = 0.04]), indicating that extensive cell death of RA mechanoreceptors resulting from an absence of RET signaling does not occur in *Gfra2* null mice. The discrepancy between E18.5 *Ret* and *Gfra2* mutant phenotypes suggests that RET signaling still occurs in neonatal *Gfra2* null RA mechanoreceptors. To demonstrate this, we quantified the expression of phospho-S6 ribosomal protein, which is downstream of RET/PI3K/mTOR signaling (*Plaza-Menacho et al., 2010*), in RA mechanoreceptors. We found that the proportion of GFP+ RA mechanoreceptors which express phospho-S6 in P0 *Gfra2^GFP/+^* control and *Gfra2^GFP/GFP^* mutant DRGs was similar (*Figure 4—figure supplement 1* [p = 0.51]). This result is consistent with the idea that RET activation occurs in neonatal RA mechanoreceptors without *Gfra2*.

Collectively, our results suggest that *Gfra2* null RA mechanoreceptors display a central projection deficit at E13.5 but recover during later development, which explains the almost normal VGLUT1 staining in layers III–V of SC at P7. In addition, our data indicate that from E15.5, an additional GFRa2 independent but RET-dependent mechanism begins to play a role in promoting the survival and central projection growth of RA mechanoreceptors.

## RET in RA mechanoreceptors is activated via both GFRa1 and GFRa2

To determine if this GFRa2-independent but RET-dependent mechanism requires GFRa1, we examined genetically labeled *Gfra1^+/−^;Ret^CreERT/+^;Rosa^Tdt^* control and *Gfra1^−/−^;Ret^CreERT/+^;Rosa^Tdt^* mutant SC and DRGs at E18.5. Similar to E13.5, neither RA mechanosensory central projections nor their number is significantly decreased in *Gfra1* null mice (*Figure 5A–D,I–J*, *Figure 4—source data 1*), suggesting that simply disrupting *trans* activation of RET via GFRa1 is not sufficient to block Ret signaling in RA mechanoreceptors.

The lack of *Ret*-mutant-like survival and central projection phenotypes of RA mechanoreceptors in both *Gfra1* and *Gfra2* single null mice made us wonder if *cis* and *trans* RET signaling function in the same developmental process and thus loss of one co-receptor can be compensated for by the other. To test this idea, we generated *Gfra1;Gfra2* double knockout mice, in which RA mechanoreceptors were specifically labeled with Tdt using the *Ret^CreERT^;Rosa^Tdt^* tandem allele. We examined control and double null SC sections and DRGs at E18.5. We found that Tdt+ RA mechanosensory central projections are greatly reduced in the dSC (*Figure 5E,G,I*, *Figure 4—source data 1*, *Gfra1;Gfra2* double mutant has $27.25 \pm 2.09\%$ of control staining at thoracic levels [p < 0.001]). In addition, fewer Tdt+ RA mechanoreceptors remain in the double knockout DRGs (*Figure 5F,H,J*, *Gfra1;Gfra2*

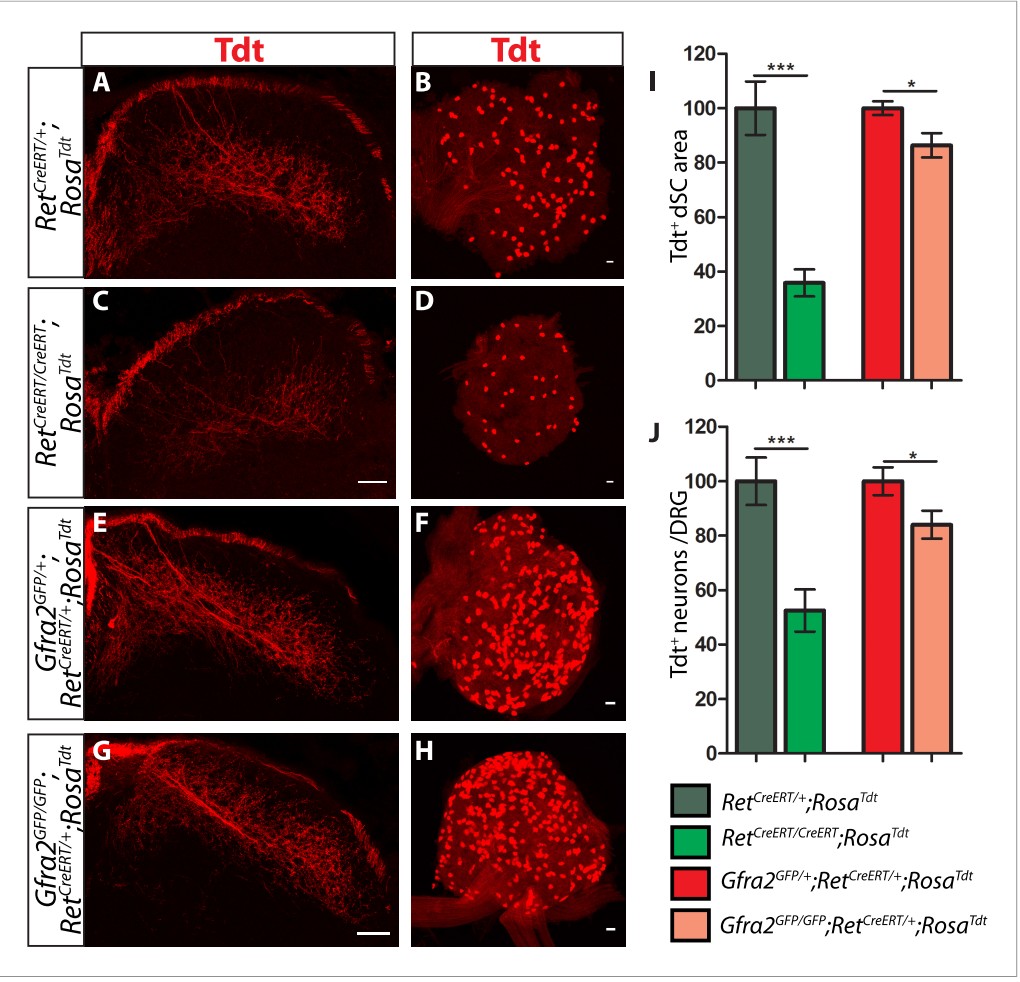

**Figure 4**. *Ret* and *Gfra2* null mice display different central projection and cell survival deficits at E18.5. (**A–H**) SC sections and whole mount L4/L5 DRGs of Tdt labeled RA mechanoreceptor from E18.5 *Ret^CreERT/+;Rosa^Tdt* control (**A–B**), *Ret^CreERT/CreERT;Rosa^Tdt* mutant (**C–D**), *Gfra2^GFP/+;Ret^CreERT/+;Rosa^Tdt* control (**E–F**), and *Gfra2^GFP/GFP;Ret^CreERT/+; Rosa^Tdt* mutant (**G–H**) embryos. (**I**) Quantification of Tdt+ pixels in dSC, which is displayed as a percentage normalized to dSC Tdt+ pixels of the within litter controls. (**J**) Quantification of the number of Tdt+ DRG neurons per whole-mount L4/L5 DRG, which is displayed as a percentage normalized to Tdt+ neurons of the within litter controls. *Ret* mutants have significant decreases in RA mechanosensory axons innervating the dSC and in the number of Tdt+ RA mechanoreceptors, suggesting that *Ret* mutants have deficits in both the growth of third order central projections and the survival of RA mechanoreceptors at E18.5. In contrast, *Gfra2* nulls have only minor deficits in RA mechanosensory central projection growth and the survival or RA mechanoreceptors, suggesting that an additional GFRa2 independent but RET-dependent mechanism functions in these processes. Scale bar = 50 μm. Error bars represent SEM. * = p < 0.05, *** = p < 0.001. Source data are provided in *Figure 4—source data 1*.

The following source data and figure supplement are available for figure 4:

**Source data 1**. RA mechanoreceptor central projections and cell number in E18.5 *Ret, Gfra2, Gfra1*, and *Gfra1;Gfra2* mutants.

**Figure supplement 1**. *Gfra2* null RA mechanoreceptors retain phospho-S6 expression.

double mutant has 38.17 ± 2.65% of control cell number [p < 0.001]), indicating that a significant number of RA mechanoreceptors die in the absence of *Gfra1* and *Gfra2*. Strikingly, the extent of reduction in both cell number and central projections of RA mechanoreceptors is comparable between the *Ret* null and *Gfra1:Gfra2* double null mice. Thus, our in vivo analyses strongly suggest that RET in RA mechanoreceptors is activated via both GFRa1 and GFRa2.

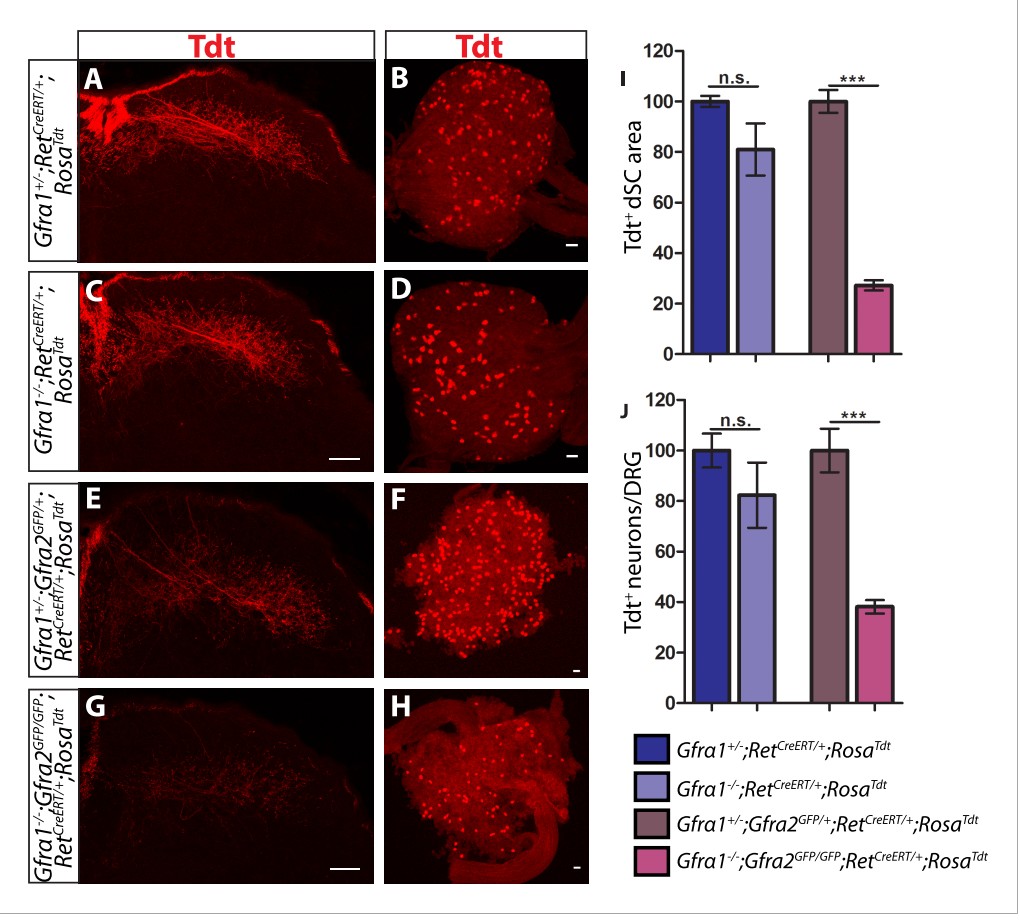

**Figure 5**. *Gfra1;Gfra2* double null mice phenocopy *Ret* mutants at E18.5. (**A–H**) SC sections and whole mount L4/L5 DRGs of Tdt labeled RA mechanoreceptors from E18.5 *Gfra1$^{+/-}$;Ret$^{CreERT/+}$;Rosa$^{Tdt}$* control (**A–B**), *Gfra1$^{-/-}$;Ret$^{CreERT/+}$; Rosa$^{Tdt}$* mutant (**C–D**), *Gfra1$^{+/-}$; Gfra2$^{GFP/+}$;Ret$^{CreERT/+}$;Rosa$^{Tdt}$* control (**E–F**) and *Gfra1$^{-/-}$; Gfra2$^{GFP/GFP}$;Ret$^{CreERT/+}$; Rosa$^{Tdt}$* double null (**G–H**) embryos. (**I**) Quantification of Tdt$^+$ pixels in dSC, which is displayed as a percentage normalized to dSC Tdt$^+$ pixels of the within litter controls. (**J**) Quantification of number of Tdt$^+$ DRG neurons per DRG, which is displayed as a percentage normalized to Tdt$^+$ neurons of the within litter controls. *Gfra1* mutants have no significant deficits in RA mechanosensory third order projections or cell survival at E18.5, indicating that ablating *trans* signaling alone is not sufficient to disrupt the development of RA mechanoreceptors. However, loss of both *cis* and *trans* signaling in *Gfra1;Gfra2* double nulls leads to a significant loss of RA mechanosensory third order projection growth and cell number, suggesting that both *cis* and *trans* RET signaling contribute to the development of RA mechanoreceptors. Scale bars = 50 µm. Error bars represent SEM. n.s. = p > 0.05, *** = p < 0.001. Source data are provided in *Figure 4—source data 1*.

## *Gfra1* is not upregulated in *Gfra2* null RA mechanoreceptors

Is RET in RA mechanoreceptors activated by GFRa1 in *cis* or *trans*? Although *Gfra1* is not widely expressed in RA mechanoreceptors in wild-type mice, could it be upregulated to compensate for the loss of *Gfra2* in the *Gfra2* null mice? To address these questions, we conducted double fluorescent in situ hybridization of *Gfra1* and *GFP* with E14.5 *Gfra2$^{GFP/+}$* control and *Gfra2$^{GFP/GFP}$* null DRG sections. We found that a comparable low number of GFP$^+$ neurons expressed *Gfra1* transcript in both mutants and controls (**Figure 6A–C** [p = 0.52]), suggesting that *Gfra1* is not upregulated in *Gfra2* null RA mechanoreceptors. In addition, we performed in situ hybridization of *Gfra1* with P0 *Gfra2$^{GFP/+}$;Ntrk1$^{+/-}$* control, *Gfra2$^{GFP/+}$;Ntrk1$^{-/-}$* null, and *Gfra2$^{GFP/GFP}$;Ntkr1$^{-/-}$* double null DRG sections. We previously showed (**Luo et al., 2009**) that *Gfra1* is expressed in NTRK1$^+$ DRG neurons and that the expression of *Gfra1* is completely lost in *Ntrk1* null mice. Here, we found that while *Gfra1* expression was observed in *Gfra2$^{GFP/+}$;Ntrk1$^{+/-}$* control DRGs, no *Gfra1* expression was observed in either *Gfra2$^{GFP/+}$;Ntkr1$^{-/-}$* null or

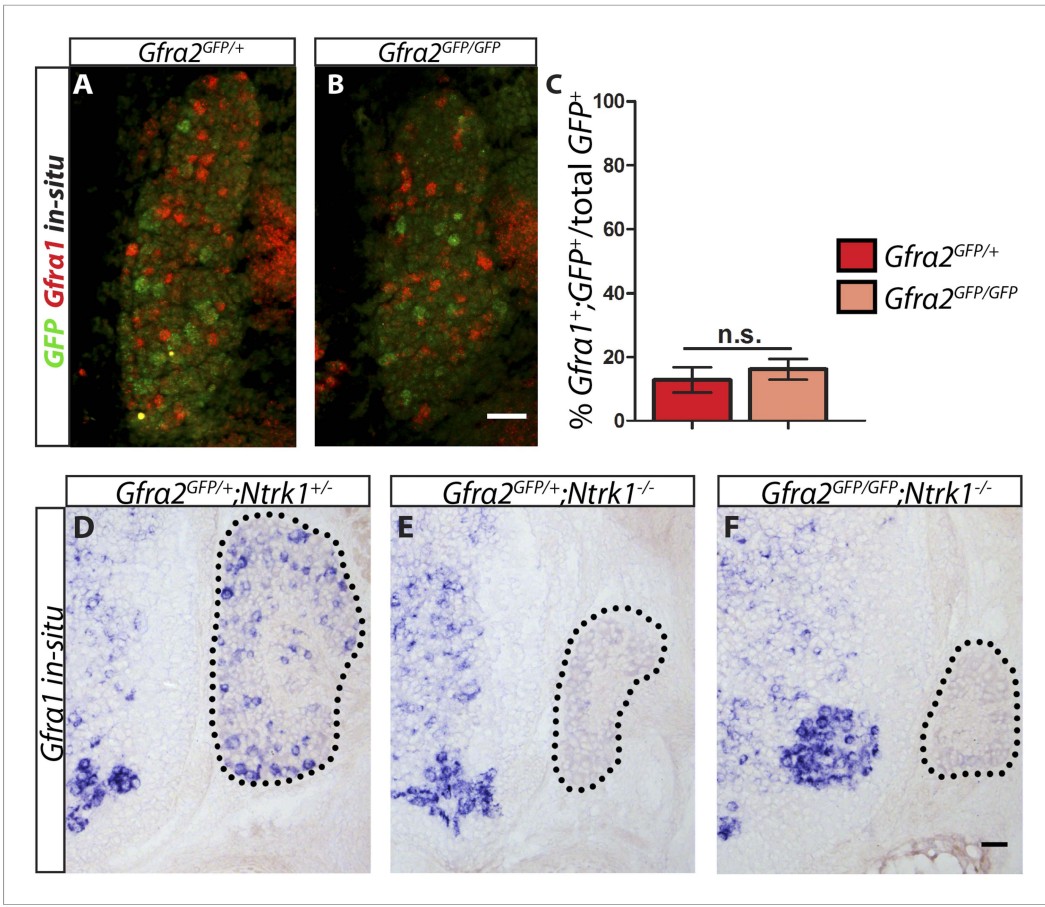

**Figure 6**. *Gfra1* is not upregulated in *Gfra2* null RA mechanoreceptors. (**A–B**) Double fluorescent in situ hybridization against *GFP* and *Gfra1* on E14.5 *Gfra2^GFP/+* control (**A**) and *Gfra2^GFP/GFP* null (**B**) DRG sections. (**C**) Quantification of percentage of *GFP^+* neurons which co-express *Gfra1*. 12.81 ± 3.92% of control *GFP^+* neurons express *Gfra1*, and 16.17 ± 3.31% of *Gfra2* null *GFP^+* neurons express *Gfra1* (p = 0.52). The comparable low number of DRG neurons co-expressing *GFP* and *Gfra1* in control and *Gfra2* nulls suggests that *Gfra1* normally is not expressed in most RA mechanoreceptors and that no upregulation of *Gfra1* occurs in *Gfra2* null RA mechanoreceptors. (**D–F**) In situ hybridization against *Gfra1* in P0 *Gfra2^GFP/+;Ntrk1^+/−* control (**D**), *Gfra2^GFP/+;Ntrk1^−/−* null (**E**), and *Gfra2^GFP/GFP;Ntrk1^−/−* double null (**F**) DRG and SC sections. Black border outlines DRG. In control DRG sections, *Gfra1* is expressed in some DRG neurons. In *Gfra2^GFP/+;Ntrk1^−/−* null DRG sections, *Gfra1* transcript is not detected because the DRG neurons which normally express detectable levels of *Gfra1* don't survive in the absence of *Ntrk1*. In *Gfra2;Ntrk1* double null mice, no *Gfra1* expression is detected in DRG neurons as well, which further supports that upregulation of *Gfra1* doesn't occur in *Gfra2* null RA mechanoreceptors. Scale bars = 50 μm. Error bars represent SEM. n.s. = p > 0.05.

The following source data and figure supplement are available for figure 6:

**Source data 1**. QPCR of *Gfra1* in embryonic *Gfra2* null DRGs.

**Figure supplement 1**. Quantitative RT-PCR (QPCR) of *Gfra1* in *Gfra2* null DRGs.

*Gfra2^GFP/GFP;Ntkr1^−/−* double null DRGs (**Figure 6D–F**). This result indicates that the expression of *Gfra1* in *Gfra2* null DRG neurons still fully depends on NTRK1 signaling and thus it must be expressed in the non-RA mechanoreceptors. Moreover, we performed quantitative RT-PCR (QPCR) for *Gfra1* transcripts in DRGs from E13.5, E15.5, and E18.5 *Gfra2^GFP/+* control and *Gfra2^GFP/GFP* mutant embryos. We found no significant difference in the expression of *Gfra1* between control and mutant DRGs at any stage (**Figure 6—figure supplement 1**, **Figure 6—source data 1**), suggesting that *Gfra1* is not transcriptionally upregulated in DRG neurons upon *Gfra2* ablation.

## GFRa1 produced by neighboring DRG neurons activates RET in RA mechanoreceptors *in trans*

Although *Gfra1* transcript in most RA mechanoreceptors is below the detection level of in situ hybridization, it remains possible that an undetectable amount of GFRa1 could function in *cis* to promote RET signaling in RA mechanoreceptors. To exclude this possibility and to demonstrate that RET in RA mechanoreceptors is indeed activated by GFRa1 in *trans*, we used DRG explants from E14.5 embryos of different mutant backgrounds and treated these explants with NRTN, GDNF, GFRa1 plus GDNF, or GFRa1 alone.

In E14.5 explants harboring the $Ret^{CFP}$ allele, the cell bodies and axons of RET$^+$ neurons, some of which are RA mechanoreceptors, can be identified by anti-GFP immunostaining. We found that CFP$^+$ neurons in $Ret^{CFP/+}$ control DRG explants grow long axons upon NRTN, GDNF, or GFRa1 plus GDNF, but not GFRa1 alone treatment (*Figure 7—figure supplement 1A–D,I*, and *Figure 7—source data 1*). In addition, the number of CFP$^+$ DRG neurons is reduced in GFRa1 alone culture (*Figure 7—figure supplement 1J*, *Figure 7—source data 2*), suggesting that either cell death or down-regulation of *Ret*, and thus *CFP* expression, occur in the absence of RET signaling. Similarly, CFP$^+$ neurons in $Ret^{CFP/CFP}$ null DRG explants lost their responsiveness to GFLs completely (*Figure 7—figure supplement 1E–H,I–J*, *Figure 7—source data 2*), suggesting that this assay reflects RET-dependent signaling.

Next, we examined DRG explants harboring the $Gfra2^{GFP}$ allele, which drives a much lower level of GFP expression than $Ret^{CFP}$. Although some small diameter DRG neurons also express *Gfra2* around P0 or later (*Luo et al., 2007*), in this $Gfra2^{GFP}$ mouse line GFP is mainly detected in the large diameter RA mechanoreceptors (*Luo et al., 2009*), which express a much higher level of *Gfra2*. Therefore, anti-GFP staining of E14.5 $Gfra2^{GFP}$ DRG explants should specifically show RA mechanoreceptors. Since GFP$^+$ axons of these explants could not be reliably imaged and quantified due to the low level of GFP expression, we approximated the extent of RET signaling in $Gfra2^{GFP}$ DRG explants by quantifying the number of discernable GFP$^+$ cell bodies. We found that $Gfra2^{GFP/+}$ control DRG neurons show robust responses upon GFL application (*Figure 7A–D,Q*, *Figure 7—source data 2*). Interestingly, $Gfra2^{GFP/GFP}$ null DRG neurons lost their responsiveness to NRTN, but retain GFP expression in the presence of either GDNF or GFRa1 plus GDNF (*Figure 7E–H,Q*, *Figure 7—source data 2*). These results suggest that a GFRa2-independent but RET-dependent mechanism can mediate GDNF responsiveness of RA mechanoreceptors.

How can *Gfra2* null RA mechanoreceptors retain their responsiveness to GDNF? It could be due to: (1) a very low level of GFRa1 is expressed in RA mechanoreceptors, which activates RET in *cis* in the presence of GDNF; or (2) GFRa1 expressed by neighboring DRG neurons binds GDNF and activates RET in RA mechanoreceptors in *trans*. To differentiate between these possibilities, we cultured E14.5 $Gfra2^{GFP/GFP}$;$Ntrk1^{-/-}$ double mutant DRGs. Since the expression of *Gfra1* in non-RA mechanoreceptor DRG neurons fully depends on NTRK1 signaling, as shown previously (*Luo et al., 2009*) and above (*Figure 6D–F*), GFRa1 should be depleted from non-RA mechanoreceptors in $Gfra2^{GFP/GFP}$;$Ntrk1^{-/-}$ double mutant DRGs. Therefore, if GFRa1 is expressed at a low level in RA mechanoreceptors and activates RET in *cis*, the double null explants should retain their responsiveness to GDNF. On the other hand, if GFRa1 expressed by neighboring neurons activates RET in RA mechanoreceptors *in trans*, the GDNF responsiveness would be lost in the *Gfra2;Ntrk1* double nulls. Here, we found that $Gfra2^{GFP/+}$;$Ntrk1^{-/-}$ control explants were responsive to NRTN, GDNF, and GDNF plus GFRa1, but not GFRa1 alone (*Figure 7I–L,R*, *Figure 7—source data 2*). In contrast, $Gfra2^{GFP/GFP}$;$Ntrk1^{-/-}$ double null explants only respond to GDNF plus GFRa1, but not to NRTN, GDNF, or GFRa1 (*Figure 7M–P,R*, *Figure 7—source data 2*). The loss of responsiveness of RA mechanoreceptors to GDNF in *Gfra2;Ntrk1* double null DRG explants strongly suggests that *Gfra1* is not expressed at a functional level in RA mechanoreceptors and that RET in *Gfra2* null RA mechanoreceptors is activated by exogenous GFRa1 from the neighboring DRG neurons in *trans*.

## GFRa1 and GFRa2 are normally shed by DRG neurons

*Trans* activation of RET could occur by direct contact between membranes of cells which express either RET or GFRa1, or by soluble GFRa1 which is shed from the cell surface. To determine whether GFRa1 is released by DRG neurons, we cultured dissociated DRGs from E18.5-P1 wild-type, $Gfra2^{-/-}$, and $Gfra1^{-/-}$ mice. We collected cell lysates and concentrated media from days 3–6 in vitro and then performed Western blot analysis.

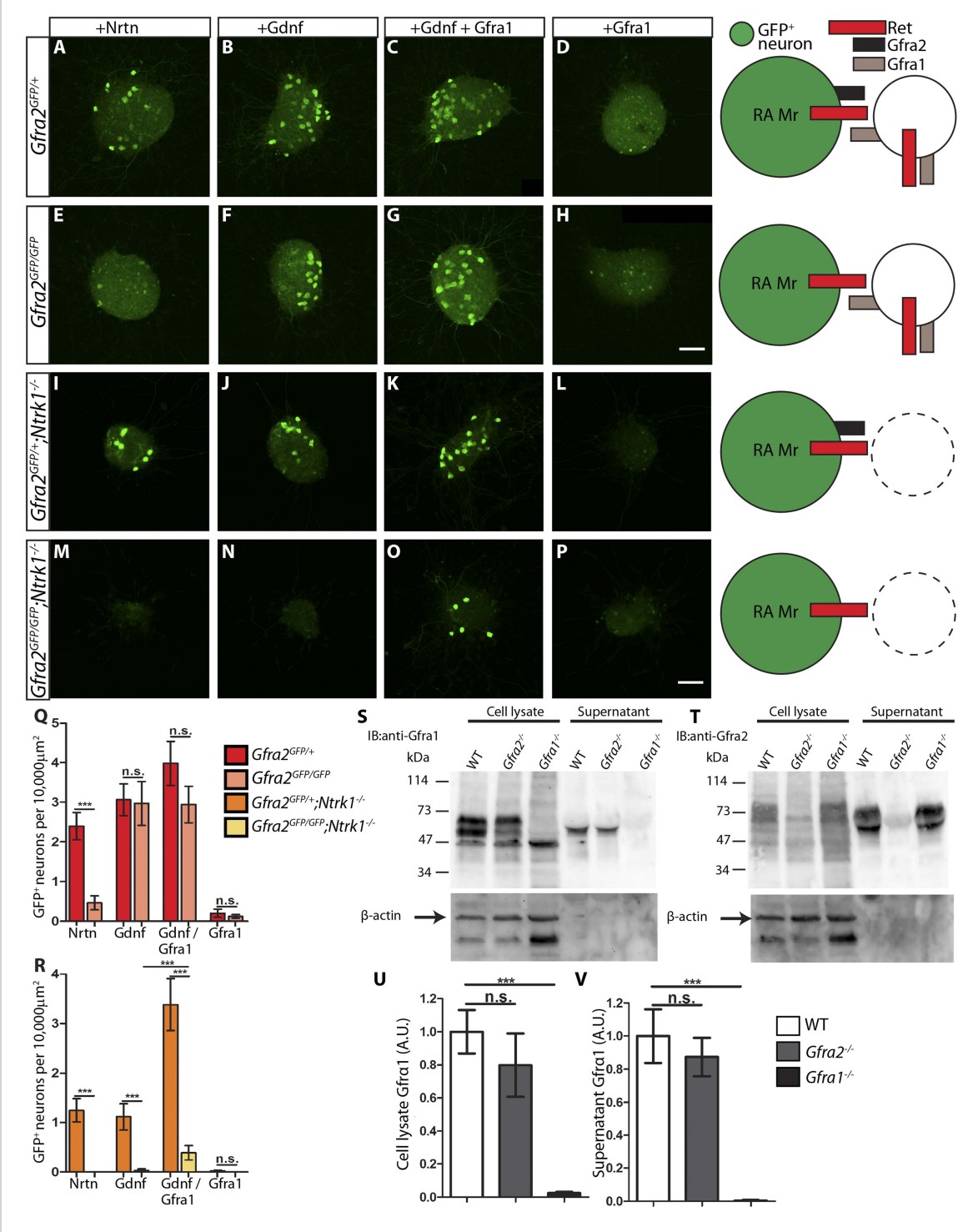

**Figure 7**. RA mechanoreceptors utilize GFRa1 produced by neighboring neurons to respond to GDNF. (**A–P**) DRG explants from *Gfra2*<sup>GFP/+</sup> control (**A–D**), *Gfra2*<sup>GFP/GFP</sup> null (**E–H**), *Gfra2*<sup>GFP/+</sup>;*Ntrk1*<sup>−/−</sup> null (**I–L**), and *Gfra2*<sup>GFP/GFP</sup>;*Ntrk1*<sup>−/−</sup> double null (**M–P**) embryos grown for 1 day in vitro and stained with anti-GFP antibody. Explants were treated with NRTN (50 ng/ml), GDNF (100 ng/ml), GDNF (100 ng/ml) plus GFRa1 (300 ng/ml), or GFRa1 (300 ng/ml), respectively. Schematic next to each genotype depicts the presence of RET and GFRas in each condition, and green color indicates cells detected by anti-GFP staining. (**Q**) Quantification of number of GFP$^+$ neurons per 10,000 μm$^2$ of explant in *Gfra2*<sup>GFP/+</sup> control and *Gfra2*<sup>GFP/GFP</sup> null explants. GFP driven from the *Gfra2* locus indicates RET signaling activity. *Gfra2* control explants display many GFP$^+$ neurons upon NRTN, GDNF, and GDNF plus GFRa1 treatment, but do not respond to GFRa1 alone. *Gfra2* null explants lose their responsiveness to NRTN, but remain responsive to GDNF and GDNF plus GFRa1. (**R**) Quantification of number of GFP$^+$ neurons per 10,000 μm$^2$ of explant in *Gfra2*<sup>GFP/+</sup>;*Ntrk1*<sup>−/−</sup> null and *Gfra2*<sup>GFP/GFP</sup>;*Ntrk1*<sup>−/−</sup> double null explants.

*Figure 7. continued on next page*

Fleming *et al*. eLife 2015;4:e06828. DOI: 10.7554/eLife.06828

*Figure 7. Continued*

In a *Ntrk1* null background, expression of *Gfra1* is lost in non-RA-mechanoreceptor DRG neurons. *Gfra2^(GFP/+)*;*Ntrk1^(−/−)* null explants respond to NRTN, GDNF, and GDNF plus GFRa1. In this case, it is likely that GDNF activates RET signaling by interacting with GFRa2 (*Jing et al., 1997*; *Sanicola et al., 1997*; *Rossi et al., 1999*). In contrast, *Gfra2;Ntrk1* double null DRG explants show GFP expression upon treatment with a combination of GDNF and GFRa1, but completely lose their responsiveness to GDNF. These results indicate that *Gfra2* null RA mechanoreceptors do not express GFRa1 at a functional level and they depend on GFRa1 produced by neighboring NTRK1+ neurons to respond to GDNF. See *Figure 7—source data 2* for quantification. (**S–V**) Western blot analysis of cell lysates and concentrated supernatants from cultured dissociated DRG neurons of E18.5-P1 wild-type, *Gfra2* null, and *Gfra1* null mice. (**S**) The specificity of the anti-GFRa1 antibody was confirmed by the loss of a doublet at the predicted size of GFRa1 in *Gfra1* null cell lysates. GFRa1 was also detected in the supernatants of wild-type and *Gfra2* null cultures, but not *Gfra1* null cultures, indicating that GFRa1 is shed from the membrane of DRGs of both wild-type and *Gfra2* mutants. Note that the size of cleaved GFRa1 is slightly smaller than that tethered to cells, which is consistent with previous publication (*Paratcha et al., 2001*). Following detection of GFRa1, membranes were stripped and probed for β-actin, which served as a loading control and confirmation that the supernatant fraction was not contaminated with cells or cellular debris (lower panel). (**T**) The specificity of the anti-GFRa2 antibody was confirmed by the loss of a band ~75 kDa in *Gfra2* null cell lysates. The larger than predicted size of GFRa2 may be due to post-translational modifications. Two GFRa2 specific bands were also detected in the supernatants of wild-type and *Gfra1* null cultures, but not *Gfra2* null cultures, indicating that GFRa2 is also shed from DRG cell membranes. The size of cleaved GFRa2 is also smaller than that tethered to cells. (**U–V**) Densimetric quantification of anti-GFRa1 blots shows no significant change in the level of GFRa1 produced by cells (**U**) or released into the media (**V**), which suggests that there is no compensation for the loss of GFRa2 through changes in the expression or release of GFRa1. See *Figure 7—source data 3* for quantification. Error bars represent SEM. Scale bars = 50 μm. n.s. = $p > 0.05$, *** = $p < 0.001$. Source data are provided in *Figure 7—source data 2, 3*.

The following source data and figure supplement are available for figure 7:

**Source data 1**. Quantification of axonal growth in *Ret* mutant DRG explants.

**Source data 2**. GFP+ neuron number in *Gfra2* null and *Gfra2;Ntrk1* double null explants.

**Source data 3**. Densimetric measurements of GFRa1 in DRG cell extracts and supernatants.

**Figure supplement 1**. *Ret^(CFP)* null DRG explants lose responsiveness to GFLs.

Immunoblotting with anti-GFRa1 revealed a doublet at ~55–65 kDa in wild-type and *Gfra2^(−/−)* cell lysates, which was absent in the *Gfra1^(−/−)* samples (*Figure 7S*, lanes 1–3), confirming the specificity of the anti-GFRa1 antibody. A positive band of ~55 kDa was present in concentrated supernatants of wild-type and *Gfra2^(−/−)* but not *Gfra1^(−/−)* cultures (*Figure 7S*, lanes 4–6), suggesting that soluble GFRa1 is shed from neonatal DRG cells. Together with reports of GFRa1 being shed by Sciatic nerve Schwann cells, immortalized neuronal progenitors (*Paratcha et al., 2001*), and adult DRG explants (*He et al., 2014*), our findings indicate that GFRa1 can be released by many cell types during both developmental and adult stages. Therefore, it is possible that RET in RA mechanoreceptors is activated in *trans* by both soluble GFRa1 and GFRa1 tethered to the membranes of neighboring cells.

In addition, although there is no significant increase of *Gfra1* transcripts in *Gfra2* null DRGs by in situ or QPCR (*Figure 6* and *Figure 6—figure supplement 1*), it remains possible that post-transcriptional regulation may occur to alter the translation, perdurance, or release of GFRa1. To test this possibility, we quantified the amount of GFRa1 in cell lysates and supernatants of wild-type and *Gfra2^(−/−)* cultures by densitometry. We found that the amount of GFRa1 expressed in the cell or shed into the media was not significantly different between wild-type and *Gfra2* null cultures (*Figure 7U–V*, *Figure 7—source data 3*). Therefore, *Gfra2* null DRGs do not produce or release more GFRa1 protein to compensate for the loss of *Gfra2*.

We also investigated whether GFRa2 is normally shed by DRGs. The specificity of the anti-GFRa2 antibody was confirmed by the absence of a ~75 kDa band from *Gfra2* null cell lysates, which was present in both wild type and *Gfra1* null cultures (*Figure 7T*, lanes 1–3). Furthermore, secreted GFRa2 band was also present in the supernatants of wild-type and *Gfra1* DRG cultures, but not in *Gfra2* null cultures. Therefore, both GFRa1 and GFRa2 are normally released by DRGs during early postnatal development.

## Dynamic expression of *Gdnf* during development

As described above, the central projections of *Gfra2* null RA mechanoreceptors display a severe, *Ret*-like deficit at E13.5, but begin to recover from E15.5, which is due to compensation by *trans*

signaling via GDNF/GFRa1. Why is *trans* RET signaling able to compensate for the loss of *cis* signaling during late embryonic development, but not at E13.5? One possible reason for the delay is the availability of *trans* signaling components. Our in situ hybridization data suggest that *Gfra1* is expressed at high levels at both E13.5 and E15.5, but the expression of *Gdnf* is greatly increased in DRGs from E13.5 to E15.5 (*Figure 1—figure supplement 1*). To provide additional evidence for the dynamic expression of *Gdnf* during development, we examined DRG and SC sections of E13.5 and E15.5 *Gdnf*[LacZ/+] (*Moore et al., 1996*) embryos using X-Gal staining. We found that the expression of LacZ increased significantly in DRGs from E13.5 to E15.5 (*Figure 8A–E* [p < 0.001], *Figure 8—figure supplement 1*). In addition, X-Gal staining was found in the E15.5 dorsal root, the pathway through which DRG central projections travel to reach the dSC (*Figure 8A–D*, black arrows). Thus, the expression of *Gdnf* seems to significantly increase in both the DRG and dorsal root from E13.5 to E15.5, providing a possible explanation for why the *trans* compensation occurs from E15.5.

## Discussion

In summary, we used RA mechanoreceptors as a model system to study the physiological functions of *trans* RET signaling and whether *cis* and *trans* activation of RET lead to the same or diversified biological outcomes in vivo. RA mechanoreceptors express *Ret* and *Gfra2* and depend on *Ret* for their survival and the growth of central axonal projections into SC, whereas *Gfra1* is highly expressed in the target field and neighboring DRG neurons. We found that the RA mechanosensory central projection deficit is negligible in postnatal *Gfra2* and *Nrtn* mutant mice. We examined central projections of genetically traced RA mechanoreceptors in *Ret*, *Gfra2*, *Nrtn*, *Gfra1*, and *Gfra1;Gfra2* double null mice and showed that only *Gfra1;Gfra2* double null mice display similar cell death and central projection deficits to those of neonatal *Ret* mutant mice, indicating that RET in RA mechanoreceptors can be activated by both GFRa1 and GFRa2. Since *Gfra1* is undetectable in control and *Gfra2* null RA mechanoreceptors, it most likely activates RET in RA mechanoreceptors *in trans*. Finally, using DRG explant cultures, we determined that *Gfra2* null RA mechanoreceptors respond to GDNF by utilizing GFRa1 produced by neighboring neurons, strongly suggesting that RET in RA mechanoreceptors is activated by GFRa1 in *trans*. Taken together, our results provide clear evidence that *cis* and *trans* RET signaling can function in the same development processes in vivo (*Figure 8F*) and that the existence of both *cis* and *trans* activation is likely to enhance but not diversify outcomes of RET signaling.

### *Trans* activation of RET in vivo

Previous expression analyses revealed that *Gfra1* is expressed more broadly than *Ret*, and cells which express *Gfra1* usually lie adjacent to *Ret*-expressing cells (*Trupp et al., 1997*; *Yu et al., 1998*). This expression pattern suggests that GFRa1 may have RET-independent functions or that GFRa1 may interact with RET expressed on the surfaces of other cells *in trans*. Indeed, evidence for both ideas has been demonstrated. GFRa1 and GDNF interact with NCAM in neurons and Schwann cells to promote neurite outgrowth and Schwann cell migration (*Paratcha et al., 2003*; *Nielsen et al., 2009*). Additionally, GFRa1 and GDNF are required for the proper migration of cortical GABAergic interneurons and can act as ligand-dependent adhesion molecules for synapse formation, independent of RET and NCAM (*Pozas and Ibáñez, 2005*; *Ledda et al., 2007*). Recently, it was also shown that GFLs have additional roles in cortical development via interactions with Syndecan-3, likely independent of RET, GFRas, and NCAM (*Bespalov et al., 2011*). On the other hand, RET can be activated by GFRa1 and GDNF in *trans* using both heterologous cells and tissue explants (*Paratcha et al., 2001*; *Ledda et al., 2002*; *Patel et al., 2012*; *He et al., 2014*). *Trans* RET signaling may affect many cellular processes, including directional axonal outgrowth and promotion of axon regeneration (*Paratcha et al., 2001*; *Airaksinen and Saarma, 2002*; *Ledda et al., 2002*).

Evidence for physiologically relevant in vivo function of *trans* RET signaling, however, has remained less conclusive. *Enomoto et al. (2004)* generated a 'cis-only' mouse model in which *Gfra1* is expressed in all RET-expressing cells, but not in cells that do not express RET. Using this model, they found that the major RET-dependent developmental processes were completely normal, suggesting that *trans* signaling is likely to be irrelevant for most RET-dependent processes. Results from this model, however, may not necessarily preclude a physiological role for *trans* signaling. Not only does this model present a loss of *trans* signaling, but it also presents a gain of function: *Gfra1* is expressed at a high level in RET[+] cells which may not normally express this co-receptor. If *cis* and *trans* RET activation lead to similar physiological outcomes, any deficits due to the loss of *trans* signaling may be masked by a gain of *cis*

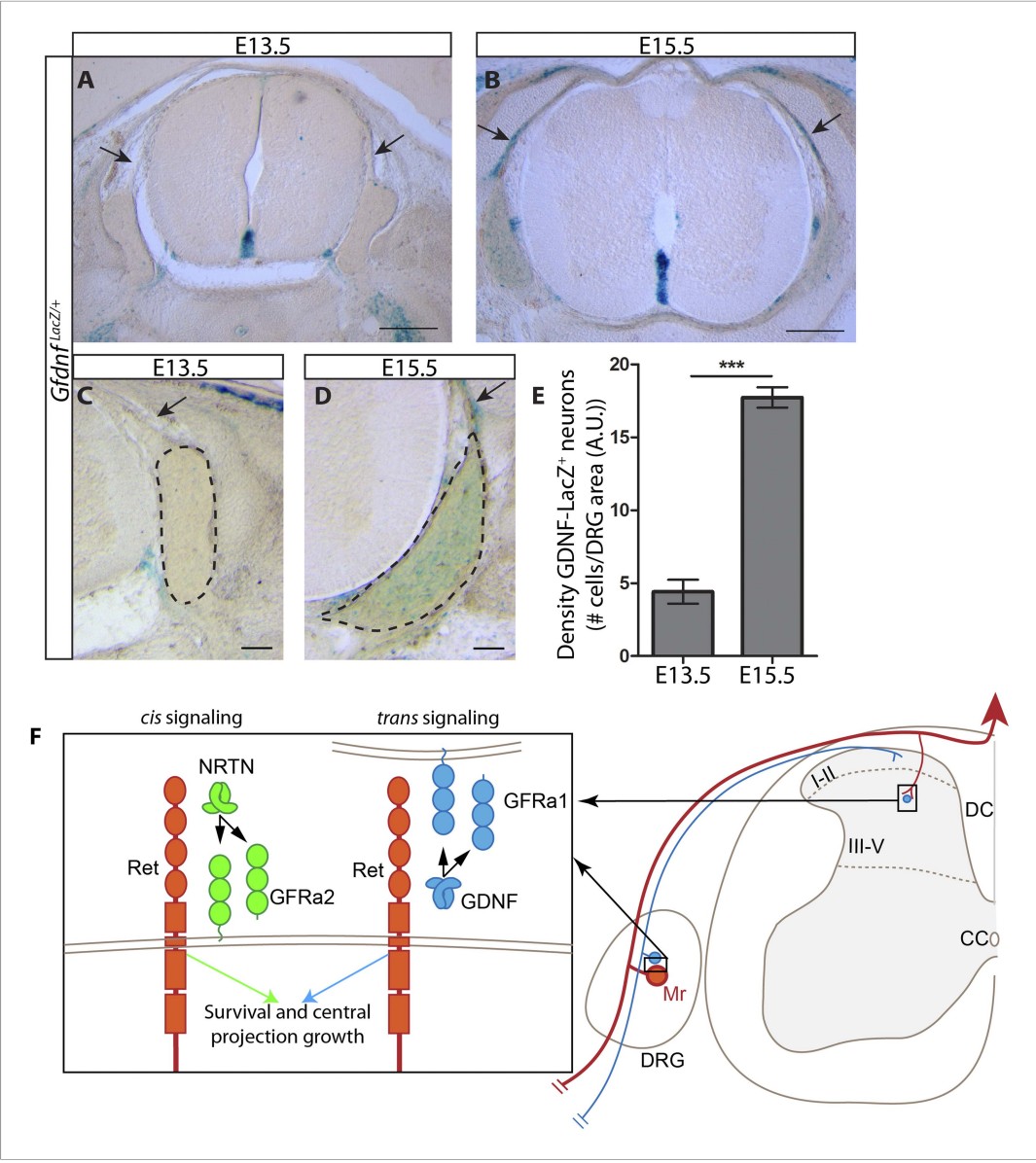

**Figure 8**. Dynamic expression of GDNF during development. (**A–D**) X-Gal staining of E13.5 (**A**, **C**) and E15.5 (**B**, **D**) *Gdnf^LacZ/+* DRG and SC sections (also see *Figure 8—figure supplement 1*). Arrows indicate dorsal roots, which express *Gdnf* at E15.5, but not E13.5. (**E**) Quantification of LacZ+ cells per DRG section, normalized to DRG area, reveals a significant increase in the number of cells expressing *Gdnf* from E13.5 to E15.5. E13.5 embryos have 4.41 ± 0.82 LacZ+ cells/unit area of DRG, E15.5 embryos have 17.73 ± 0.70 LacZ+ cells/unit area of DRG (p < 0.001). Error bars represent SEM. Scale bars = 200 μm (**A–B**), 100 μm (**C–D**). *** = p < 0.001 (**F**) Model of *cis* and *trans* signaling at cell bodies and central branches of RA mechanoreceptors. GFRa2 is co-expressed with RET in RA mechanoreceptors and can activate RET in *cis*. GFRa2 can also be shed from the membrane and may activate RET in its soluble form. GFRa1 is expressed in neighboring DRG neurons, dorsal root entry zone cells, and dSC cells. GFRa1 present at the membrane of these cells may directly contact the cell bodies or processes of RA mechanoreceptors to activate RET in *trans*. In addition, soluble GFRa1 released from these cells may also activate RET in RA mechanoreceptor in *trans*.

The following figure supplement is available for figure 8:

**Figure supplement 1**. *Gdnf^LacZ* expression in DRGs at E13.5 and E15.5.

signaling. Indeed, the gain of function of GFRa1 was recently demonstrated in enteric hematopoietic cells derived from *cis*-only mice (*Patel et al., 2012*). Thus, whether *trans* RET signaling has any physiological function in development has remained an open question.

We aimed to address this question by analyzing the survival and growth of RA mechanosensory central projections using loss-of-function mouse lines. Here, we found that loss of *cis* signaling, via ablation of either *Gfra2* or *Nrtn*, produces a central projection deficit during early embryonic development (*Figure 2*). Our findings are consistent with previous observations using a different *Ret* knock-in line and NRTN ectopic expression (*Honma et al., 2010*), but differ from the findings using anti-GFRa2 staining to visualize RA mechanosensory central projections at E13.5 (*Bourane et al., 2009*). This is likely due to different subcellular localization of CFP and GFRa2, as well as the expression of GFRa2 by some dSC cells (*Figure 1—figure supplement 1E,F*), which may mask the RA mechanoreceptor phenotype. Interestingly, this phenotype recovers during late embryonic development and the central projections of *Gfra2* null mice seem nearly normal postnatally (*Figures 1, 4*). Thus, our results suggest that *cis* RET signaling is required for the initial growth of RA mechanosensory central projections, but an additional *cis* signaling independent process takes place during later development. Indeed, the loss of both *cis* and *trans* signaling in *Gfra1;Gfra2* double mutants recapitulates the *Ret* phenotype (*Figures 5, 6*). Furthermore, using DRG explant and dissociated culture, we demonstrated that soluble GFRa1 is normally released by DRGs and that GFRa1 produced by NTRK1$^+$ DRG neurons present a potential source to activate RET in RA mechanoreceptors in *trans* to promote their survival (*Figure 7*). Taken together, our results suggest that *trans* RET signaling contributes to the development of RA mechanoreceptors in vivo.

Nevertheless, the exact subcellular locus of *trans* RET activation in RA mechanoreceptors remains speculative. The expression pattern of *Gfra1* suggests that *trans* RET activation is possible in the axons of RA mechanoreceptors along their path to dSC, and/or at the cell body within the DRG. Although individual DRG cell bodies are surrounded by satellite glial cells, large macromolecules and proteins are able to invade the space between the neuron and satellite cell (*Hanani, 2005*), suggesting *trans* RET activation by soluble GFRa1 could occur within DRGs.

## RET signaling and the survival of RA mechanoreceptors

Signaling of neurotrophic RTKs, such as NTRK1, NTRK2, and NTRK3, is critical for the specification and survival of numerous classes of neurons (*Ernsberger, 2009*). RET signaling also plays important roles in survival, differentiation, and specification of distinct neuronal classes (*Enomoto, 2005*). For example, RET signaling components are absolutely required for the survival of enteric neurons (*Taraviras et al., 1999*), but their roles in DRG neuron survival are more complicated to dissect. Previously, it was reported that the number of total DRG neurons is not significantly reduced in neonatal and early postnatal *Ret* mutants (*Luo et al., 2007*). At that time, specific molecular markers or genetic approaches for labeling RA mechanoreceptors had not been identified, so it was impossible to specifically assay the role of RET signaling in the survival of this neuronal population. Given that RA mechanoreceptors represent a very small proportion of the total DRG neurons (*Molliver et al., 1997*; *Luo et al., 2007*), a partial loss of this population may not lead to a significant change in cell counts of total DRG neurons.

In another paper (*Luo et al., 2009*), it was proposed that RA mechanoreceptors depend on NRTN-GFRa2/Ret signaling for their development. This was based on the findings that GFRa2 is the only co-receptor expressed in RA mechanoreceptors and that *Ret*, *Gfra2*, and *Nrtn* null mice display the same no-Pacinian-corpuscle phenotype. Since RET can't be used as a molecular marker to quantify the number of RA mechanoreceptors in *Ret* null mice, the number of P0 RET$^+$/NTRK1$^-$ and *Gfra2*$^{GFP}$ DRG neurons, most of which indicate the RA mechanoreceptors, was quantified in *Nrtn* and *Gfra2* nulls. No significant change in cell number between mutants and controls was found, suggesting that *Gfra2* and *Nrtn* are not required for survival of neonatal RA mechanoreceptors. These results are interesting in light of the current findings. Here, we show that when *cis* signaling via NRTN/GFRa2 is perturbed (as was tested in [*Luo et al., 2009*]), *trans* signaling via GDNF/GFRa1 can activate RET in RA mechanoreceptors to support their survival and central projection growth. When the number of genetically labeled RA mechanoreceptors was quantified in different mutant backgrounds, we found only marginal changes in *Gfra2* nulls but drastic decreases in *Ret* mutants at E18.5 (*Figure 4*). The slight difference between the current and previous findings regarding the loss

of RA mechanoreceptors in *Gfra2* nulls at P0 (*Luo et al., 2009*) is likely due to different methods in identifying and quantifying RA mechanoreceptors. In short, the current study clarifies that RET signaling is required for RA mechanoreceptor survival but simple disruption of *cis* RET signaling components may not reveal this deficit.

## Co-existence of *cis* and *trans* RET signaling

What is the purpose for RET to be activated both in *cis* and in *trans*? Do *cis* and *trans* signaling activate different cellular responses and influence distinct developmental processes, or do *cis* and *trans* signaling exert similar physiological effect? Here, we found that, in RA mechanoreceptors, *cis* and *trans* signaling seem to produce similar biological outputs in vivo. Our results demonstrate that *cis* and *trans* signaling can compensate for the loss of each other to promote both the central projection growth and survival of RA mechanoreceptors (*Figure 8F*). This compensatory ability suggests that the existence of both *cis* and *trans* activation is likely to enhance but not diversify outcomes of RET signaling. Consistent with this notion, a recent study found that peripheral nerves secrete both GDNF and GFRa1, which attracts perineural invasion of heterogeneous cancer cells, some of which expresses *Ret* and *Gfras*, while some express only *Ret* (*He et al., 2014*).

In an attempt to show that GFRa1 is normally released from wild-type DRG cells for *trans* RET signaling, we found the same for GFRa2 (*Figure 7*). Given that soluble GFRa2 could also activate RET in *trans* with NRTN (*Worley et al., 2000*), our finding raises many interesting questions, such as whether all GFRas are secreted and whether '*cis*' and '*trans*' RET signaling normally co-exist even when RET and GFRas are expressed in the same cells. It seems plausible that even for GFRas co-expressed with RET (usually defined as '*cis*' signaling), such as GFRa2 in RA mechanoreceptors, it could be secreted and then upon NRTN binding activates RET in the cell from which it was released ('*trans*' activation).

Although *cis* and *trans* activation of RET lead to a similar biological outcome in the growth and survival of RA mechanoreceptors, it is worth noting that substantial differences likely exist between the signaling processes of *cis* and *trans* RET activation. *Cis* RET activation might be more efficient, given that GFRas and RET are located in the same membrane. In addition, *cis* and *trans* signaling could differ in the kinetics of recruitment of RET to lipid rafts upon GFLs stimulation, interactions with downstream-associated proteins, and the longevity of activated RET and downstream effectors (*Tansey et al., 2000*; *Paratcha et al., 2001*). Additionally, it remains possible that although the gross structure of the RA mechanosensory central projections seems mostly normal in mice lacking either *cis* or *trans* signaling, more precise aspects of circuit formation, such as specific synapse formation, which are beyond the resolution of current analysis, may differentially depend on *cis* or *trans* signaling. Finally, it will be interesting to see whether *cis* and *trans* signaling can produce similar biological outcomes in other systems, as we have shown here for RA mechanoreceptors. Future experiments to carefully dissect *cis* and *trans* RET signaling in other types of cells and tissues will address these issues.

# Materials and methods

## Mouse strains

Mice except *GDNF^lacZ* line were raised in a barrier facility in Hill Pavilion, the University of Pennsylvania. All procedures were conducted according to animal protocols approved by Institutional Animal Care and Use Committee (IACUC) of the University of Pennsylvania and National Institutes of Health guidelines. *GDNF^lacZ* mice were raised in accordance with the European Community Council Directive of November 24, 1986 (86/609/EEC), and approved by the ethics. Most mice used in this paper were described previously: *Ret^CreERT*, *Ret^CFP*, *Nrtn^{+/−}* (purchased from the Jackson lab), *Gfra2^GFP* (re-derived using sperm provided by Dr Jeffery Milbrandt at the Washington University), *Rosa26^Tdt*, and *Gdnf^LacZ* mice (*Moore et al., 1996*; *Heuckeroth et al., 1999*; *McDonagh et al., 2007*; *Uesaka et al., 2008*; *Luo et al., 2009*; *Madisen et al., 2010*). The *Ntrk1^−* allele was generated by crossing the floxed *TrkA^F592A* allele (*Chen et al., 2005*) to germline Cre mice. The generation of *Gfra1* conditional and null mice and the *Ret^CreERT*;*Rosa^Tdt* tandem allele are described below. All mice except *Gdnf^LacZ* were maintained on a mixed C57BL/6J and CD1 background. *Gdnf^LacZ* mice were maintained on a C57BL/6N background. Except for *Gfra1*;*Gfra2* double null animals (n = 2), at least three animals per genotype were examined. N-values for explants are listed in *Figure 7—source data 1, 2*.

## Generation of *Gfra1* conditional and null alleles

We generated *Gfra1* conditional knockout mice, in which loxP sites flank exon 6 of *Gfra1*, by homologous recombination. Mice harboring the floxed allele were crossed to germ line Cre mice, resulting in a *Gfra1* allele lacking exon 6. The loss of *Gfra1* transcript in *Gfra1*$^{-/-}$ mice was confirmed by in situ hybridization of mutant and control DRGs (see *Figure 1—figure supplement 3*).

## Generation of *Ret*$^{CreERT}$;*Rosa*$^{Tdt}$ tandem allele

Since *Ret* and *Rosa26* loci are located only ~5 megabases apart on mouse chromosome 6, we generated a tandem configuration of *Ret*$^{CreERT}$ and *Rosa*$^{Tdt}$ alleles, which are linked during meiosis and became a great genetic advantage for our experiments (*Figure 3—figure supplement 1*). We used this tandem *Ret*$^{CreERT}$;*Rosa*$^{Tdt}$ allele to specifically label RA mechanoreceptors in different mutant mouse lines described in the text.

## Genetic labeling of RA mechanoreceptors

We set up timed pregnancy mating for mice in the evening and checked mice for vaginal plugs the following morning. The time when a female mouse was found to have a plug was counted as embryonic day 0.5 (E0.5). We treated embryos harboring the *Ret*$^{CreERT}$;*Rosa*$^{Tdt}$ reporter allele with 4-hydroxy-tamoxifen (4-HT, 2 mg and 1 mg at E11.5, and E12.5, respectively) by oral gavage to pregnant female mice to specifically label RA mechanoreceptor population.

## Tissue preparation and histology

Spinal columns of embryos and neonatal mice at the desired developmental stages were dissected out and directly immersed in PBS/4% paraformaldehyde (PFA) for 2 to 4 hr at 4°C. Postnatal mice were sacrificed with $CO_2$, transcardially perfused with 4% PFA, and spinal columns were dissected out and post-fixed with 4% PFA for 2 hr at 4°C. They were then cryo-protected in 1× PBS, 30% sucrose overnight. 20-μm frozen sections of SC and DRGs were cut using a CM1950 cryostat (Leica, Buffalo Grove, IL). Immunostaining of SCs and DRG sections were performed as described previously (*Niu et al., 2013*). Antibodies used are as follows: rabbit anti-GFP (A-11122, 1:2000, Invitrogen, Carlsbad, CA), chicken anti-GFP (GFP-1020, 1:1000, Aves, Tigard, OR), chicken anti-NF200 (NF-H, 1:500, Aves), rabbit anti-NF200 (N4142, 1:1000, Sigma, St. Louis, MO), rabbit anti-cRet (18121, 1:50, IBL, Minneapolis, MN), rabbit anti-NTRK1 (06-574, 1:1000, Millipore, Temecula, CA), guinea pig anti-VGLUT1 (AB5905, 1:1000, Millipore), rabbit anti-phospho-S6 (2215s, 1:200, Cell Signaling, Beverly, MA), and Alexa Fluorescent conjugated Goat or Donkey secondary antibodies (1:500, Invitrogen or Jackson ImmunoResearch, West Grove, PA).

## LacZ color reaction

Embryos of the desired age were eviscerated and fixed in 1% PFA, 2 mM $MgCl_2$, 5 mM EGTA, 0.02% NP40, and 0.2% glutaraldehyde in phosphate buffer (pH 7.4) for 1.5 to 2 hr at 4°C. Vibratome sections were incubated for 30 min in washing solution (2 mM $MgCl_2$, 0.02% NP-40 in phosphate buffer pH 7.4). LacZ reaction was developed with X-gal (1 mg/ml) at 37°C.

## In situ hybridization

DIG- or FITC-labeled riboprobes were synthesized using a DIG or FITC RNA labeling kit (11175025910, Roche, Indianapolis, IN). Template for GFP was amplified by PCR and subcloned into vector pGEM-T Easy (A1360, Promega, Madison, WI). Antisense RNA probes for *Ret, Gfra1, Gfra2, Gdnf,* and *Nrtn* were generated as previously described (*Luo et al., 2009*). The detailed procedures of in situ hybridization and double fluorescent in situ hybridization were performed as described previously (*Fleming et al., 2012*).

## QPCR

DRGs from E13.5, E15.5, and E18.5 *Gfra2*$^{GFP/+}$ and *Gfra2*$^{GFP/GFP}$ embryos were dissected and rapidly frozen on dry ice. RNA was extracted with the GeneJet RNA Purification Kit (K0731, Fermentas, Vilnius, Lithuania) and cDNAs were generated using Super-Script III First-Strand Synthesis System (18080-51, Invitrogen). 500 ng of total RNA was used for each RT reaction in a total volume of 25 μl. QPCR reactions were performed in triplicate for three samples of each age and genotype. QPCR reactions contained

SYBR Green PCR master mix (4309155, Life Technologies, Carlsbad, CA), 0.5 µM of each primer, and 3 µl (for *Gfra1*) or 1 µl (for *Gapdh*) of cDNA template per 15 µl reaction. Reactions were run and analyzed on a StepOnePlus Real-Time PCR System (Applied Biosystems, Carlsbad, CA). Primers used were *Gapdh* (5′-CCACCAACTGCTTAGCCCCC-3′ and 5′-GCAGTGATGGCATGGACTGTGG-3′) and *Gfra1* (5′-TGTCTTTCTGATAATGATTACGGA-3′ and 5′-CTACGATGTTTCTGCCAATGATA-3′). p-values between samples were calculated from ΔCT values with the Student's t-test, and relative concentrations were calculated by the $2^{-\Delta\Delta CT}$ method (*Livak and Schmittgen, 2001*).

## DRG explant culture and immunostaining

E14.5 embryos were removed from the dam and placed in F-12 media (11765-047, Invitrogen) on ice. SCs with attached DRGs were dissected from the spinal column, and individual DRGs were removed and placed in fresh F-12 on ice. Using a dissecting needle, DRGs were cleaned and bisected, and then placed in fresh F-12. Explants were grown on Superfrost Plus slides (22-034-979, Fisher, Waltham, MA) coated with poly-L-lysine (P1274, Sigma, 0.1 mg/ml in ddH2O overnight at 4°C) and laminin (354232, BD, Franklin Lakes, NJ), 20 µg/ml in HBSS [14170122, Invitrogen] at 37°C for one to 3 hr). Immediately before placing explants on the slide, slides were washed with HBSS and DRG culture medium (Neurobasal medium [21103-049, Invitrogen], 1× B27 [17504-044, Invitrogen], 100 U/ml penicillin/streptomycin [15140-122, Invitrogen], 2 mM L-glutamine [25030-081, Invitrogen], and 35 mM glucose). DRG culture media supplemented with the appropriate recombinant proteins (50 ng/ml Nrtn [477-MN-025, R&D, Minneapolis, MN], 100 ng/ml GDNF [512-GF-010, R&D], 300 ng/ml GFRa1 [560-GF-100, R&D], or 100 ng/ml GDNF and 300 ng/ml GFRa1) were added to the culture dish. Four to six DRG explants were placed on each slide and the culture dishes were carefully moved to a 37°C incubator and left undisturbed overnight. Following 16–24 hr of incubation, cultures were rinsed with PBS and fixed in 4% PFA in PBS for 30 min at room temperature. Immuno-fluorescence was then performed directly in the culture dish using antibody dilutions described above. Following secondary antibody, explant slides were mounted on microscope slides using Superglue, and coverslipped with Fluoromount-G (0100-01, Southern Biotech, Birmingham, AL) and 22 × 22 mm coverglass.

## Dissociated DRG cultures and biochemistry

DRGs from E18.5-P1 mice were collected into HBSS on ice. DRGs were first digested in 0.5 mg/ml collagenase (LS4186, Worthington, Lakewood, NJ) plus 100 U/ml penicillin/streptomycin, 10 mM HEPES, and 1× MEM vitamins (M6895, Sigma) in MEM (11095072, Invitrogen) at 37°C for 30 min, followed by a second digestion with 0.05% trypsin (25200056, Invitrogen) plus 100 U/ml penicillin/streptomycin, 10 mM HEPES, and 1× MEM vitamins in MEM at 37°C for 30 min. Digestion was stopped by adding 5% FBS and 10 mM HEPES in HBSS. Cells were then triturated with a fire polished Pasteur pipette to a homogenous solution. The cells were then pelleted at 500×*g* for 5 min and resuspended in DRG culture media, as described above, supplemented with 50 ng/ml NRTN, 100 ng/ml GDNF, and 50 ng/ml NGF (556-NG-100, R&D). Cells were plated in six-well collagen-coated plates (Millipore, PICL06P05) and cultured at 37°C and 5% CO₂. After 2 days, media were removed and cells were rinsed with warmed Neurobasal media. 2 ml of fresh DRG culture media supplemented with NRTN, GDNF, and NGF (but without B27) was added to each well. After 2 days, media were removed and saved at 4°C with added protease inhibitors (P8340, Sigma). Fresh media supplemented with growth factors but without B27 were then added to each well. After an additional 2 days, media were removed and pooled with previously collected media, and additional protease inhibitor was added. The cells were rinsed twice with PBS, and then lysed directly in the well by the addition of 70 µl 2× sample buffer (0.125 M Tris pH 6.8, 20% glycerol, 4% SDS, 0.16% bromophenol blue, 10% 2-mercaptoethanol) and scraping, followed by heating at 95°C for 5 min. All cell lysates were then brought to a total volume of 140 µl with 1× PBS. Supernatants were centrifuged at 14,000×*g* for 15 min to clear cellular debris, and then were concentrated to ~30 µl with Amicon 30 kDa filters (UFC503024, Millipore), then mixed with an equal volume of 2× sample buffer and heated at 95°C for 5 min.

Duplicate 4–15% gradient mini-Protean TGX gels (456-1084, Biorad, Hercules, CA) were used to run samples. 40 µl of cell lysate of each genotype or one third of the total volume of concentrated supernatant of each genotype was used. Both gels were then transferred to nitrocellulose membrane and blocked in 3% BSA in TBS plus 0.1% Tween-20 (TBST) for 1 hr at room temperature. Membranes were then incubated overnight with either goat anti-GFRa1 (0.2 µg/ml, AF560, R&D) or goat

anti-GFRa2 (0.2 µg/ml, AF613, R&D) in blocking solution overnight at 4℃. Following washes with TBST, membranes were incubated with donkey anti-goat-AP (SC-2022, 1:5000, Santa Cruz Biotechnology, Santa Cruz, CA) in blocking solution for 1 hr at room temperature. After washes, AP was detected with CDP-Star (T2218, Applied Biosystems) and membranes were imaged with a Chemi-Doc system (BioRad).

Following imaging, membranes were stripped with 2× 10 min stripping buffer (0.2 M glycine, 0.1% SDS, 1% Tween-20, pH 2.2), followed by 2× 10 min wash with PBS and 2× 5 min wash with TBST. Membrane was then probed with rabbit anti-β-actin (sc-130656, 1:400, Santa Cruz Biotechnology) and goat anti-rabbit-AP (T2191, 1:5000, Applied Biosystems) following the above procedure, except that all blocking and antibody incubations were performed in 5% milk in TBST.

Western blot densitometry was performed with ImageJ. Three cultures per genotype were analyzed. Densitometry measurements for each antibody were performed on three blots running independent culture samples. Relative quantifications were performed using β-actin in the cell lysates as a measure of total protein per lane, and optical density values for GFRa1 were scaled accordingly. Because an equal proportion of total lysates was run in each lane, total β-actin per cell lysate lane was used as a proxy for cell number, and was therefore used to normalize protein levels in the supernatant lanes (equal proportion of total supernatant volume were run in each lane). Cell lysate and supernatant samples were scaled to wild-type quantifications of respective sample type and reported in arbitrary units. Student's t-test was used to measure significance of differences between genotypes.

## Image acquisition

Fluorescent images of SC/DRG sections were acquired on a Leica SP5II confocal microscope. DRG explant cultures were imaged on Leica DM5000B microscope. Bright field images were taken using Leica DM5000B microscope.

## Quantification and Statistics

For histological analysis, at least six sections per specified spinal/DRG level per animal were analyzed. For quantification of genetically labeled neuron number in E18.5 embryos, whole-mount L4/L5 DRGs were imaged and total Tdt+ cell number was counted in each DRG. Except for *Gfra1;Gfra2* double null animals (n = 2), at least three animals per genotype were examined. N-values for explants are listed in *Figure 7—source data 1* and *2*. Pixel counts for central projections were generated by counting the number of pixels at each intensity level (0–256) in an outlined immunoreactive area in ImageJ. Background staining was subtracted by counting pixel number of each intensity level in a non-immunoreactive region of the tissue. The minimal intensity level at which two consecutive levels displayed a pixel count of zero was taken as the threshold cut of background fluorescence. Pixel counts of real staining were then calculated by summing the pixel counts for all intensity levels above the defined background level. Column graphs were generated in GraphPad Prism 5. All error bars are ± standard error of the mean (SEM), unless otherwise specified. All statistical analyses were performed using SAS version 9.3 (SAS Inc., Cary, NC). Due to differences in labeling efficiency across litters in 4-HT treated mice, quantification for SC section staining and whole mount DRGs were performed by normalizing to controls within the same litter. For all explant quantifications, GFP+ neuron number per 10,000 µm² was calculated for each explant. For *Ret^CFP* explants, a circle with a radius 200 µm larger than the explant was drawn around the explant in ImageJ, and the number of CFP+ axons which crossed the circle was counted.

## Acknowledgements

We thank Drs G Bashaw, DD Ginty, M Ma, B Pierchala, and other Luo lab members for helpful comments on the manuscript. We thank Ming Lu for statistical consultation. WL is supported by National Institutes of Health (NIH, R00NS069799 and 1R01NS083702), the Basil O'Connor Starter Scholar Research Award (Grant No. 5-FY12-109 from the March of Dimes Foundation), and the Klingenstein-Simons Fellowship Awards in the Neurosciences. MF is supported by NIH (F31NS086168, T32HD007516, and T32GM07517) and the Hearst Foundation. Work in the Klein lab is supported by the Max-Planck Society.

# Additional information

## Funding

| Funder | Grant reference | Author |
| --- | --- | --- |
| National Institutes of Health (NIH) | R00NS069799 | Wenqin Luo |
| March of Dimes Foundation | Basil O'Connor Starter Scholar Research Award, 5-FY12-109 | Wenqin Luo |
| Klingenstein Third Generation Foundation (KTGF) | Klingenstein-Simons Fellowship Awards in the Neurosciences | Wenqin Luo |
| Max-Planck-Gesellschaft | Research Grant | Sónia Paixão, Rüdiger Klein |
| National Institutes of Health (NIH) | F31NS086168 | Michael S Fleming |
| Hearst Foundations | Student Fellowship | Michael S Fleming |
| National Institutes of Health (NIH) | 1R01NS083702 | Wenqin Luo |
| National Institutes of Health (NIH) | T32HD007516 | Michael S Fleming |
| National Institutes of Health (NIH) | T32GM07517 | Michael S Fleming |

The funders had no role in study design, data collection and interpretation, or the decision to submit the work for publication.

## Author contributions

MSF, SP, WL, Conception and design, Acquisition of data, Analysis and interpretation of data, Drafting or revising the article; AV, JN, Acquisition of data, Drafting or revising the article; RK, Conception and design, Analysis and interpretation of data, Drafting or revising the article; JMS, Drafting or revising the article, Contributed unpublished essential data or reagents

## Ethics

Animal experimentation: Mice except GDNFlacZ line were raised in a barrier facility in Hill Pavilion, the 618 University of Pennsylvania. All procedures were conducted according to animal protocols 619 approved by Institutional Animal Care and Use Committee (IACUC) of the University of 620 Pennsylvania and National Institutes of Health guidelines. GDNFlacZ mice were raised in 621 accordance with the European Community Council Directive of November 24, 1986622 (86/609/EEC), and approved by the ethics.

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
