## [Decision Letter]

Thank you for sending your work entitled “*Cis* and *trans* RET signaling control the survival and central projection growth of rapidly adapting mechanoreceptors” for consideration at *eLife*. Your article has been favorably evaluated by a Senior editor and two reviewers, one of whom is a member of our Board of Reviewing Editors.

The Reviewing editor and the other reviewer discussed their comments before we reached this decision, and the Reviewing editor has assembled the following comments to help you prepare a revised submission.

Both reviewers felt that this work is comprehensive and addresses an important question in development neurobiology. Both reviewers agreed that the conclusions are strongly supported by the experiments. They conclude that this manuscript is potentially suitable for publication in *eLife* after the authors address the following questions:

1) It is unclear why anti-VGLUT1 staining (green) was not interfered by intrinsic GFP in *Gfrα2*^*GFP*^ allele (Figure 1).

2) The authors used a germline Cre to delete *Gfrα1* gene. So this is not a really conditional knockout. This information is hard to find (in supplemental method) and should be stated in the main text. Why not to use DRG specific *cre*?

3) DRG neuronal cell bodies are enveloped by satellite glias. How does the *trans* cRet signaling occur between RA and other neurons where satellite glias are present in between? Can diffusible GFRαs penetrate satellite glias or are cell bodies the primary sites for *trans* signaling? Maybe RA axons are the primary sites. These possibilities should be discussed.

4) Figure 1—figure supplement 1 is not very informative since most of the expression data were published from previous studies. In my opinion, the important information here is to show that RET and *Gfrα1* are not expressed in the same type of cells. From this figure, I cannot draw that conclusion.

5) In Figure 1, why do the *Gfrα1* homozygous mice show more branches compared with control?

6) The authors showed convincingly that *Gfrα1* can be shed and can act in *trans*. Is it possible that the membrane tethered *Gfrα1* act in vivo to execute this function?

---

## [Author Response]

*1) It is unclear why anti-VGLUT1 staining (green) was not interfered by intrinsic GFP in* Gfrα2^GFP^
*allele (*Figure 1*)*.

The level of GFP driven by this *Gfrα2*^*GFP*^ allele is somehow low. The GFP expressed in GFRα2^+^ DRG neurons is not directly visible in fixed tissue, and can only be visualized by anti-GFP antibodies. Additionally, based on our own experience and communication with Dr. Jeff Mildbrant, whose lab originally generated the line, anti-GFP antibody can only reveal cell bodies of some GFRα2^+^ DRG but not axons. This point is demonstrated in the explant experiments in Figure 7. Finally, the punctate anti-VGLUT1 staining is very different in appearance from the axonal staining revealed by the *Ret*^*CFP*^ and the *Ret*^*CreERT*^*;Rosa*^*tdTomato*^ alleles, further suggesting that the staining observed in Figure 1 is not axonal staining of the RA mechanoreceptor central projections.

To make this clear, we have added the following text to the legend of Figure 1: “Note that GFP driven from the *Gfrα2* locus cannot be visualized directly. Therefore, positive green fluorescent staining indicates presynaptic VGLUT1^+^ puncta but not GFRα2^+^ primary afferent axons.”

*2) The authors used a germline Cre to delete* Gfrα1 *gene. So this is not a really conditional knockout. This information is hard to find (in supplemental method) and should be stated in the main text. Why not to use DRG specific* cre*?*

We first generated the conditional allele, and then generated the null allele by crossing to a germline *cre* (*Sox2*^*Cre*^). In the experiments presented in this manuscript, however, we used only the null allele. To make this clearer, we removed the mention of the conditional allele in the subsection headed “Central projections of RA mechanoreceptors are normal in E13.5 *GFRα1* null mice”. To accurately describe our methods, we still describe the generation of the conditional and null alleles in Figure 1—figure supplement 3 and in the supplemental experimental procedures.

We did not use a DRG specific Cre because there are many potential sources of *trans-*expressed GFRα1 besides neighboring DRG neurons, such as non-neuronal cells in the DRG, cells in the dorsal root, and cells in the dorsal spinal cord. Therefore, using a DRG neuron specific Cre would not eliminate all potential sources of *trans* activating GFRα1.

*3) DRG neuronal cell bodies are enveloped by satellite glias. How does the* trans *cRet signaling occur between RA and other neurons where satellite glias are present in between? Can diffusible GFRαs penetrate satellite glias or are cell bodies the primary sites for* trans *signaling? Maybe RA axons are the primary sites. These possibilities should be discussed*.

We have added the following sentences to the Discussion to address these issues: “Nevertheless, the exact subcellular locus of *trans* RET activation […] suggesting *trans* RET activation by soluble GFRα1 could occur within DRGs.”

*4)*
Figure 1—figure supplement 1
*is not very informative since most of the expression data were published from previous studies. In my opinion, the important information here is to show that RET and Gfrα1 are not expressed in the same type of cells. From this figure, I cannot draw that conclusion*.

We agree that the expression patterns of *Ret*, *Gfrα1*, and *Gfrα2* in spinal cord and DRG have previously been described (as indicated by the citations in the main text). However, we feel having the data at these specific developmental timepoints readily available in the supplemental information is helpful for readers who may not be familiar with or remember all previous characterizations. The in-situ hybridization data for *Gdnf* and *Nrtn* at these timepoints has not previously been described, and therefore this is novel information.

Just to clarify, RET and GFRα1 are co-expressed in some DRG neurons at E13.5, but they are distinct from those DRG neurons co-expressing RET and GFRα2. The RET/ GFRα1 positive neurons completely depend on *TrkA* for their survival and/or expression of *Ret* and *Gfrα1.* In contrast, RET/ GFRα2 positive neurons are independent of *TrkA*. These data were described in our previous publication (Luo, et al., Neurons, 2009, Figure 1), as we cited in the first paragraph of the Results section. Thus, we didn’t repeat these experiments but summarized the previous findings in Figure 1—figure supplement 1. Consistently, our double in situ hybridization of *Gfrα1* and *GFP* (driven from *Gfrα2* locus) with E14.5 WT DRG neurons in Figure 6 showed little overlap between these two populations of neurons.

*5) In*
Figure 1*, why do the Gfrα1 homozygous mice show more branches compared with control?*

Based on our unpublished observations, dorsal spinal cord neurons precociously express *Ret* (and therefore *CFP*) in *Gfrα1* null mice at E13.5. At present, we don’t know whether this is due to gene regulation or abnormal migration of spinal cord neurons (normally there are RET+ neurons in the middle and ventral spinal cord at E13.5). Since we quantified total axonal growth of RA mechanoreceptors by measuring CFP^+^ pixel number in the dorsal spinal cord, CFP+ axons/dendrites of these precocious RET+ dorsal spinal cord neurons are also included in the quantification of *Gfrα1* null mice, which likely accounts for the increased dorsal spinal cord CFP+ pixels. We have added this information to the legend of Figure 1: the increased CFP signal in *Gfrα1* null dSC is likely due to the precocious expression of *Ret* in some dSC neurons of *Gfrα1* mutants.

*6) The authors showed convincingly that Gfrα1 can be shed and can act in* trans*. Is it possible that the membrane tethered Gfrα1 act in vivo to execute this function?*

Although GFRα1 is shed by DRG cells, a large proportion of GFRα1 likely remains membrane tethered. Therefore, it is likely that both the soluble and membrane-tethered pools could contribute to *trans* RET activation. To make this point clear, we have added the following sentence to the subsection headed “GFRα1 and GFRα2 are normally shed by DRG neurons”: “Therefore, it is possible that RET in RA mechanoreceptors is activated in *trans* by both soluble GFRα1 and GFRα1 tethered to the membranes of neighboring cells.” Additionally, our model in Figure 8 shows that both membranes, tethered and soluble GFRα1, are likely to activate RET in *trans*.